# Forty new genomes shed light on sexual reproduction and the origin of tetraploidy in Microsporidia

Amjad Khalaf[1]*, Chenxi Zhou[1,2], Claudia C. Weber[1], Emmelien Vancaester[1],
Ying Sims[1], Alex Makunin[1], Thomas C. Mathers[1], Dominic E. Absolon[1],
Jonathan M. D. Wood[1], Shane A. McCarthy[1], Kamil S. Jaron[1], Mark Blaxter[1],
Mara K. N. Lawniczak[1]

1 Tree of Life, Wellcome Sanger Institute, Wellcome Genome Campus, Cambridge, United Kingdom,
2 Department of Genetics, University of Cambridge, Cambridge, United Kingdom

* ak37@sanger.ac.uk

## Abstract

Microsporidia are single-celled, obligately intracellular parasites with growing public health, agricultural, and economic importance. Despite this, Microsporidia remain relatively enigmatic, with many aspects of their biology and evolution unexplored. Key questions include whether Microsporidia undergo sexual reproduction, and the nature of the relationship between tetraploid and diploid lineages. While few high-quality microsporidian genomes currently exist to help answer such questions, large-scale biodiversity genomics initiatives, such as the Darwin Tree of Life project, can generate high-quality genome assemblies for microsporidian parasites when sequencing infected host species. Here, we present 40 new microsporidian genome assemblies from infected arthropod hosts that were sequenced to create reference genomes. Out of the 40, 32 are complete genomes, eight of which are chromosome-level, and eight are partial microsporidian genomes. We characterized 14 of these as polyploid and five as diploid. We found that tetraploid genome haplotypes are consistent with autopolyploidy, in that they coalesce more recently than species, and that they likely recombine. Within some genomes, we found large-scale rearrangements between the homeologous genomes. We also observed a high rate of rearrangement between genomes from different microsporidian groups, and a striking tolerance for segmental duplications. Analysis of chromatin conformation capture (Hi-C) data indicated that tetraploid genomes are likely organized into two diploid units, similar to dikaryotic cells in fungi, with evidence of recombination within and between units. Together, our results provide evidence for the existence of a sexual cycle in Microsporidia, and suggest a model for the microsporidian lifecycle that mirrors fungal reproduction.

**Data availability statement:** All relevant data and Supporting information are available in full on Zenodo https://doi.org/10.5281/zenodo.17251512. Small supporting figures and tables were attached to submission as well.

**Funding:** This work was funded in whole by the Wellcome Trust (grant number 220540/Z/20/A). Funding was awarded to authors MKNL and MB. The funders did NOT play any role in the study design, data collection and analysis, decision to publish, or preparation of the manuscript. Wellcome Trust URL: https://wellcome.org/.

**Competing interests:** The authors have declared that no competing interests exist.

**Abbreviations :** ANI, Average Nucleotide Identity; BUSCO, Benchmarking Using Single Copy Orthologues; DToL, Darwin Tree of Life; OTU, operational taxonomic unit; ToLIDs, Tree of Life identifiers; WGD, whole genome duplication.

## Introduction

Biology is characterized by intimate interactions and a fundamental interdependence between organisms living in close proximity. Yet, our understanding of symbionts and cobionts (organisms sampled alongside a target organism) typically lags behind our understanding of their hosts. Primarily, this is due to difficulties in accessing an organism's symbionts, growing them, and/or assessing their behavior. One such example is Microsporidia, which are single-celled, spore-forming, obligately intracellular parasites [1]. They were first described as the agent of a disease known as "pébrine" in farmed silkworms (*Bombyx mori*, Lepidoptera), which had caused crises in the industry along the silk road in the late 1800s [2,3]. Since then, Microsporidia have been identified in many different hosts, and are now appreciated as parasites of a broad range of metazoans, and even some protozoans [4,5]. Nine genera have been identified as human pathogens, especially in immunocompromised individuals, causing a range of symptoms such as diarrhea, encephalitis, keratitis, sinusitis, and myositis [6–10]. Similarly, arthropod-infecting microsporidians constitute a growing problem for beekeeping and aquaculture industries worldwide [11–14]. Other microsporidians are being explored as potential malaria transmission control agents, with evidence associating infection of *Anopheles* mosquitoes to a reduction in *Plasmodium* transmission [15–17].

Despite their importance, microsporidian biology remains fairly enigmatic, owing in part to their obligately intracellular nature, small size, low prevalence in most host populations, and biological quirks such as cryptomitosis (where the condensation and separation of chromatin into distinct chromosomes are unclear) [4,18,19]. One oddity that some microsporidian species share with diplomonads such as *Giardia intestinalis* is the persistence of two equivalent nuclei inside one cell, closely appressed against each other with distinct nuclear membranes and synchronous replication, for the whole lifecycle—a form known as a "diplokaryon" [20–23]. Some microsporidians remain monokaryotic for all their lifecycle [24], and others cycle between diplokaryotic and monokaryotic forms [21,25–28].

The cycling between monokaryotic and diplokaryotic states in some microsporidian species has widely been assumed to be part of a meiotic reproductive cycle, and phenomena interpreted as gametogenesis, plasmogamy, karyogamy, and synaptonemal complexes have been reported [21,24,29–40]. Whilst strictly diplokaryotic and strictly monokaryotic species were assumed to only undergo asexual mitosis [41], recent observations such as a very transient monokaryotic stage and synaptonemal complexes in diplokaryotic species suggest otherwise [42,43]. Population-level genetic data has also suggested the occurrence of recombination events in both monokaryotic and diplokaryotic species [44–50]. Although the presence of meiosis has never been validated beyond morphological data, homologs of genes involved in meiosis have been identified in many microsporidian species, with phylogenetic analysis indicating that all Microsporidia may have descended from a sexual fungal ancestor [18].

Furthermore, the ploidy of each microsporidian nucleus, and whether that changes during developmental or meiotic cycles, remains an open question. After the first

discovery of polyploidy in Microsporidia, tetraploidy has been suggested to manifest in the diplokaryotic form [51], with species proposed to cycle between haploid-diploid or diploid-tetraploid states as part of their reproductive cycles [52]. Some doubt has been cast on this now that seven microsporidian species have been reported to be tetraploid, including both monokaryotic and diplokaryotic species [51,53,54]. Additionally, some microsporidians have species-specific lifecycle variants, such as the formation of an octet of spores as one infective unit (an "octospore") [29,55], and it is unknown whether the ploidy of these is the same as their monokaryotic/diplokaryotic forms.

Ploidies other than haploid and diploid are common in nature but generally phylogenetically unstable, and are usually resolved back to effective diploidy through loss of one set of chromosomes or genome-wide rediploidisation leaving the signatures of whole genome duplication [56]. The high frequency of apparent tetraploids in Microsporidia [53] could be a reflection of a tendency towards formation of tetraploids or a reflection of a fundamental tetraploid state. It is unclear whether observed microsporidian polyploids arose through autopolyploidy, where the four genome copies derive from within-species events, or allopolyploidy, where genomes from different species are combined through hybridization (reviewed in [57]). Furthermore, it is unknown whether tetraploidy in Microsporidia is characterized by a single ancient event that has been stably maintained across multiple lineages, an outcome that would be highly unusual given that rediploidisation is typically a key process that follows whole-genome duplications [58–66], or whether polyploidy has arisen independently in each microsporidian lineage where it is observed.

While the number of microsporidian genome assemblies is increasing, few are high-quality, chromosome-level assemblies. High-quality assemblies are crucial to disentangling questions about the occurrence of sexual reproduction, the origins of polyploidy, and the interaction of the two [67]. Large-scale reference genome and next-generation sequencing initiatives, such as the Darwin Tree of Life (DToL) [68], can incidentally generate high-quality genome assemblies for cobionts when sequencing host species, and thus offer an unrivaled data-generation opportunity for rare and unculturable endosymbionts. For instance, DToL has recently released over 100 novel *Wolbachia* genomes and two cnidarian endoparasite genomes assembled from data arising from individual hosts targeted for reference genomes [69,70]. Some screens for Microsporidia have also been carried out on DToL data, yielding a few microsporidian genome sequences [67,71,72].

Here, we present microsporidian genomic data from 40 host organisms sequenced at the Wellcome Sanger Institute as part of DToL. We recover eight partial and 32 complete (or nearly complete, with Benchmarking Using Single Copy Orthologues [BUSCO] completeness scores >70%) microsporidian genomes. Eight of our complete genomes are chromosome-level assemblies, seven of which, to our knowledge, were scaffolded with the first Hi-C data generated for Microsporidia. These new genomes represent much of the breadth of currently described microsporidian diversity, with genomes from five of the seven microsporidian clades named by Bojko and colleagues [4]. We show that one tetraploid genome is organized into two units, likely the nuclei of the diplokaryon, but also show evidence of historical recombination between all four genomes. We describe rearrangements between the haplotypes of some genomes, chromosomal rearrangements in microsporidian evolution, and a high tolerance for segmental duplications. We recognize recombination signatures in other tetraploid genomes, and propose a model to synthesize our observations of ploidy, reproduction, and the microsporidian lifecycle.

## Results

### Forty new microsporidian genome assemblies

We identified host organisms carrying microsporidian infections by screening the raw genome sequencing data generated by DToL for microsporidian sequences, or by PCR amplification of microsporidian targets from host DNA extracts before genome sequencing. We also observed microsporidian-like sequences in additional genomes that proved to be horizontal DNA transfers into the host genome in the absence of current live infections. From 40 host species, we recovered 32 complete microsporidian genome assemblies with BUSCO completeness score >70%, including eight chromosome-level genomes, seven of which were scaffolded with Hi-C data. We also assembled eight partial genome sequences, with

BUSCO completeness score <70%. The recovered microsporidian genomes come from hosts belonging to eight insect orders, with lepidopteran and dipteran hosts yielding 13 and 12 microsporidian genomes, respectively (Fig 1). The hosts include several not previously known to be infected by microsporidia, including *Loensia variegata* (Psocodea, bark lice), *Vulgichneumon bimaculatus* (Hymenoptera, ichneumon wasp), and *Delia platura* (Diptera, seedcorn maggot fly) (S1 Table). Our sample size is small, but we note that most microsporidian genomes from lepidopteran hosts are derived from the microsporidian group Nosematida (77%, $n = 10$), and most microsporidian genomes from dipteran hosts are derived from Amblyosporida (67%, $n = 8$). While some microsporidians are known to distort the sex of their host populations [74], we found no evidence of such in our limited sampling (S1 Fig). The full list of hosts, their recovered microsporidian genome assemblies, and associated genome summary statistics are given in S1 Table. See Materials and methods for steps taken to confirm that the microsporidian genomes come from single species and not mixed infections.

The genome sequences were placed in five of the seven microsporidian clades identified by Bojko and colleagues [4]. Half derived from Nosematida (two chromosome-level genomes, 13 complete genomes, five partial genomes) and 11 from Amblyosporida (six chromosome-level genomes, five complete genomes) (Figs 1 and 2). Four belonged to Entero-cytozoonida (one chromosome-level genome, two complete genomes, one partial genome), two to Neopereziida (two partial genomes), and three to Glugeida (three complete genomes) (Figs 1 and 2). Most chromosome-level and complete genome assemblies presented here have comparable or higher contiguity and BUSCO completeness compared to previously published microsporidian genome assemblies (Fig 2). Our complete genome sequences range in span from 2.35 Mb to 56 Mb. The eight largest genome assemblies are unpurged due to their read depth being too low to run purge_dups [80], or because of the presence of severe between-haplotype rearrangements, or due to the lack of Hi-C data; and thus

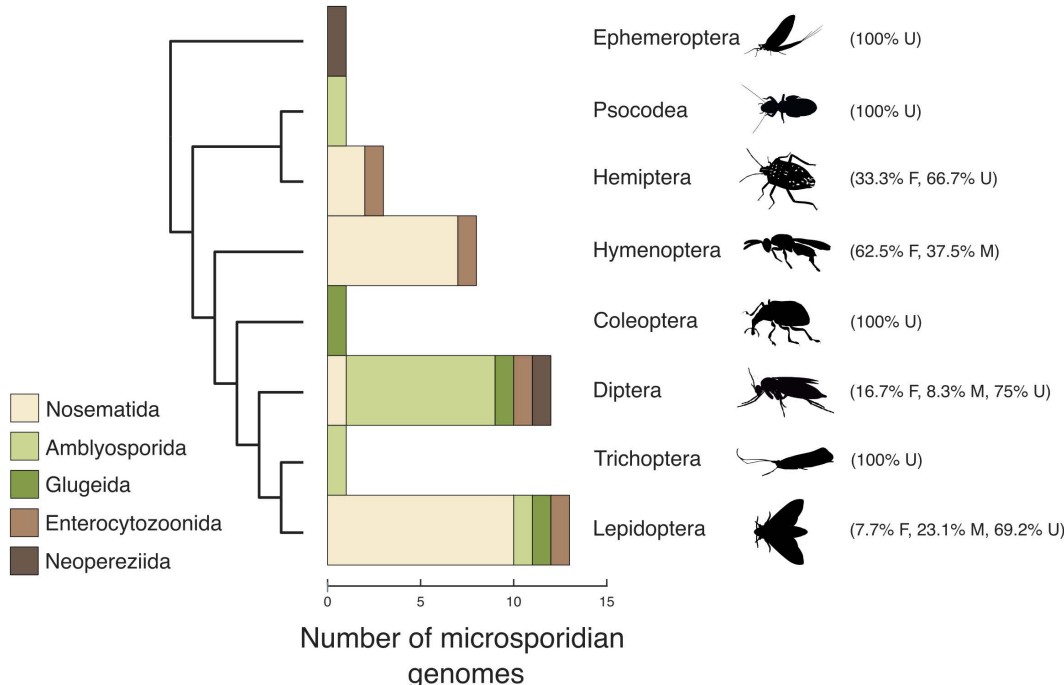

**Fig 1. Prevalence of Microsporidia in DToL insect genomes.** Microsporidian genomes recovered from insect hosts, split by taxonomic order. F: female, M: male, U: unspecified sex. The silhouettes used in this figure were taken from https://www.phylopic.org, and are all under CC0 1.0 Universal Public Domain Dedication. Credits: Ephemeroptera, Nathan Jay Baker; Psocodea, Christina N. Hodson; Hemiptera, Dave Angelini; Hymenoptera, Emma Kärrnäs; Coleoptera, Kanako Bessho-Uehara; Diptera, Christina N. Hodson; Trichoptera, Christoph Schomburg; and Lepidoptera, Andy Wilson. The data underlying this figure can be found in S1 Table. The figure was generated using ToyTree [73], and manually annotated using InkScape (version 1.2.2).

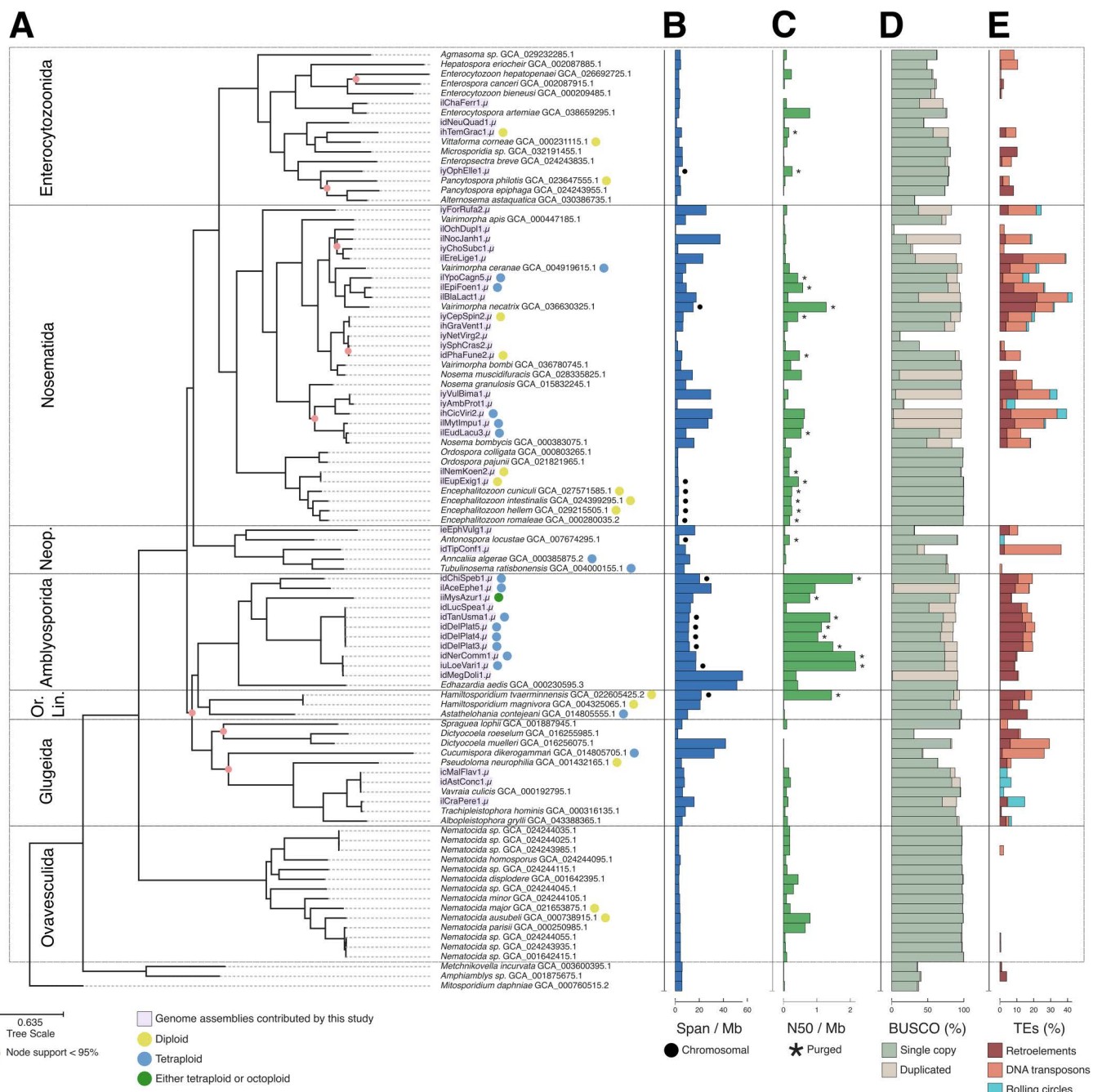

**Fig 2. 600 gene phylogeny of Microsporidia. (A)** ASTRAL [75] phylogeny summarizing individual phylogenies of 600 BUSCO genes (microsporidia_odb10) [76] across all publicly available microsporidian genome assemblies (excluding multiple strains where they are available), and the genome assemblies generated in this study (*n* = 40, marked in purple). The full phylogeny with all publicly available genomes, including different strains, is found in S3 Fig. Branch lengths were estimated with IQ-TREE using a concatenated alignment of the individual BUSCOs [77]. Nodes with less than 95% support are marked with pink circles. Ploidy is marked in circles at the tips of the tree for genomes where it was characterizable. **(B)** Genome assembly span (Mb) as calculated by assembly-stats (Github: https://github.com/sanger-pathogens/assembly-stats), with black circles marking chromosome-level genome assemblies. **(C)** N50 values (Mb) as calculated by assembly-stats (Github: https://github.com/sanger-pathogens/assembly-stats), with asterisks marking purged genome assemblies. **(D)** BUSCO gene (microsporidia_odb10) completeness percentage, marked in green for single-copy genes, and beige for duplicated genes. **(E)** Transposable element percentage as predicted by RepeatModeler and RepeatMasker [78,79], marked in burgundy for retroelements, peach for DNA transposons, and blue for rolling circles. Neop.: Neopereziida; Or. Lin.: Orphan Lineage. The data underlying (A) can be found in S1 Text. The data underlying (B), (C), (D), and (E) can be found in S1 Table. The figure was generated using ToyTree [73], and manually annotated using InkScape (version 1.2.2).

retain haplotypic duplication (which also results in the high BUSCO duplication scores observed in Fig 2). The largest purged assembly (idChiSpeb1.μ, see Materials and methods for an explanation of the naming system used) is 20 Mb (Fig 2). This falls within the range of previously sequenced microsporidian genomes, which range in size from 2.2 Mb (*Encephalitozoon romalae*) to 51.3 Mb (*Edhazardia aedis*) [81].

Ploidy was estimated using GenomeScope2 and Smudgeplot [82], following criteria we outlined previously [53]. In brief, we estimated ploidy for genomes with average read depth exceeding 20× (see Materials and methods for full explanation). As a result, we were able to characterize 14 of our genomes as polyploid, and five as diploid (Fig 2). We note that some microsporidian genomes had large segments differing in copy number from the majority of the genome. These occurred both within and between chromosomes. The duplications varied in length, and preferentially occurred towards the ends of contigs or chromosomes (see S2 Fig for details on how these were identified and examples). Such cases are common, with some level of segmental duplication observed in nearly all of the 14 polyploid genomes (see File Collection 1 at https://doi.org/10.5281/zenodo.17251512). No other phylogenetic or host metadata unites those genomes.

We also performed repeat annotation (with RepeatModeler and RepeatMasker) on all genomes [78,79] so that all genomes were annotated congruently. To assess the relationship between the phylogeny and transposable element loads, and genome spans, we computed transformations representing the fit of each feature with the tree's topology ($\lambda$), branch-lengths ($\kappa$), and root–tip distance ($\delta$) [83] (S3 Table). For the majority of examined traits, we found no significant phylogenetic signal, with the exception of a strong root–tip distance effect on all transposable element loads. Retroelement load and DNA transposon loads were moderately correlated to one another and genome span, whereas helitron load was weakly correlated to all other features (S4 Table). We also explored the distribution of transposable elements and rRNA sequences along chromosome-level genomes (File Collections 2 and 3 at https://doi.org/10.5281/zenodo.17251512). Whilst in Encephalitozoons transposable elements and rRNAs clustered in subtelomeric regions [84], the majority of chromosomes from other genomes did not exhibit a similar pattern.

The phylogenetic tree suggests a revision of microsporidian relationships. We confidently placed Nosematida as a sister group to Enterocytozoonida, and the group known as 'orphan lineage' (containing the genera *Hamiltosporidium* and *Astathelohania*) as a sister group to Amblyosporida, in agreement with previous whole-genome phylogenies [4,85,86]. However, with moderate confidence (node with 87% support, see newick string in S1 Text), we place Glugeida a sister group to the ancestor of the orphan lineage and Amblyosporida. The previously contested place of Neopereziida as a sister group to the ancestor of Nosematida and Enterocytozoonida was confidently confirmed (Fig 2).

The classification of species traditionally assigned to *Nosema*, such as *Nosema* (=*Vairimorpha*) *ceranae* and *Nosema* (=*Vairimorpha*) *apis*, has been the subject of ongoing debate [4,54,87–89]. Our phylogeny, using 600 loci, strongly supports the split between *Vairimorpha* and *Nosema*, with *Nosema* (=*Vairimorpha*) *ceranae* clustering robustly with other *Vairimorpha* genomes (Fig 2).

## Operational taxonomic unit (OTU) classification of species suggests autotetraploidy in Microsporidia

Morphological and histopathological data are usually employed to identify microsporidia to species level, but no such data were available for these newly assembled microsporidian genomes. We measured phylogenetic branch lengths between every possible combination of two genomes to establish a proxy baseline for genomic disparity among individuals that are known to be members of the same species based on morphological, histopathological, or cell culture data. We then used this baseline to assess whether any of the new genomes could be diagnosed as members of known species, and whether the homeologous subgenomes of our tetraploid genomes could be characterized as belonging to the same species (autotetraploidy) or not (allotetraploidy). We note that morphological and histopathological data remain important for species delineation, and such approaches may not be applicable to organisms where the traditional species concept may not apply.

Publicly available genomes classified to the same species exhibited high genomic similarity, with an average branch length distance of 0.01 amino acid substitutions per site (using the BUSCO-based phylogeny, see S3 Fig). On the other hand, genomes from different species were much more divergent, with an average branch length of 0.577 amino acid substitutions per site. One pair of species, *Hamiltosporidium tvaerminnensis* and *Hamiltosporidium magnivora*, had particularly short branch lengths (0.012–0.017 amino acid substitutions per site). Excluding these, the smallest branch length between different species within the same genus was 0.15 amino acid substitutions per site.

Relying on these data, we established a conservative branch length threshold to classify genomes as belonging to the same species based on the smallest distance between *H. tvaerminnensis* and *H. magnivora* genomes (0.012 amino acid substitutions per site). Using this criterion, we classified 17 of the new genomes as belonging to a known species or an unnamed species formed by two or more of our genomes (S6 Table). If within-species divergence was permitted to extend to the full range observed within species (i.e., to 0.032 substitutions per site), one additional new genome was classified as being a likely member of a named species. From these analyses, we identified five species-like groupings of otherwise unidentified microsporidia sequenced here (S6 Table). Those include gmOTU1 comprising iuLoeVari1.µ (from host *Loensia variegata* [Psocodea]), idNerComm1.µ (from host *Neria commutata* [Diptera]), and idMegDoli1.µ (from host *Megamerina dolium* [Diptera]); and gmOTU2 from five dipteran hosts: idDelPlat3.µ (from host *Delia platura* [Diptera]), idTanUsma1.µ (from host *Tanytarsus usmaensis*), idDelPlat4.µ (from host *Delia platura*), idLucSpea1.µ (from host Lucilla sp.), and idDelPlat5.µ (from host *Delia platura*). For an explanation of the naming system used for OTUs, see Materials and methods.

In this context, we also explored within-individual BUSCO divergences between homeologous subgenomes within tetraploid assemblies. We found that in all but one case at least 85% of homeologous gene pairs were separated by distances smaller than the relaxed threshold (0.032 substitutions per site) used for species delineation (Fig 3), consistent with autotetraploidy. The 15% of more divergent homeologous gene pairs were scattered across contigs in the assembly, suggesting they are isolated cases of increased divergence (Oxford dot plots in File Collection 4 at https://doi.org/10.5281/zenodo.17251512). In the remaining case, ilAceEphe1.µ (from host *Acentria ephemerella* [Lepidoptera]), over 40% of homeologous gene pairs exceeded the same-species divergence threshold (Fig 3A) and many of these divergent homeologues were segregated on separate contigs (Fig 3B). We note that while we are unable to distinguish between gene copies arising from polyploidisation events *versus* other segmental duplication events for any genome, this high proportion of divergent homeologues in ilAceEphe1.µ may indicate it is the product of a recent hybridization event between two related diploid individuals (i.e., an allotetraploid). Furthermore, this pattern in ilAceEphe1.µ is unlikely to be the result of a mixed infection of two diploid individuals as the read depths of the four subgenomes are congruent with them belonging to one single genome (File Collection 1 at https://doi.org/10.5281/zenodo.17251512).

## Homeologous genomes coalesce more recently than do species in most tetraploid Microsporidia

With the discovery of tetraploidy in many microsporidian lineages [51, 53, 54], one of the key questions is whether tetraploidy is an ancient event shared between multiple (or all) tetraploid microsporidian lineages and subsequently lost by diploid lineages nested within ancestrally tetraploid clades, or is phylogenetically recent and the result of multiple, independent lineage-specific events. The generation of resolved tetraploid genome assemblies allows us to address this question. In the absence of recombination, gene conversion, and sexual reproduction, under ancient tetraploidy we should find that the two homeologous genomes within a tetraploid species have a coalescence deeper than that of the homologous genomes compared between related species [93]. However, if recombination among homeologous genomes occurs, the homeologous genomes within an individual may on average coalesce more recently than they do between species, even while some loci retain a signal of the deep coalescence of the lineages that contributed to the tetraploid. On the other hand, if tetraploidy is the product of recent, independent events, haplotypes will coalesce more recently than species, even in the absence of recombination, and there will be no deep coalescence signal.

**A**

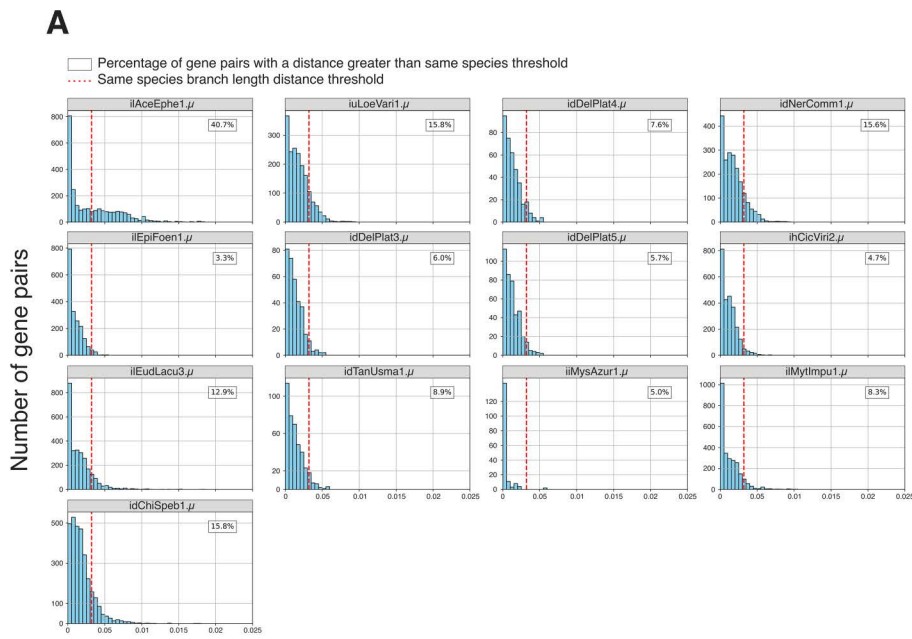

**B**

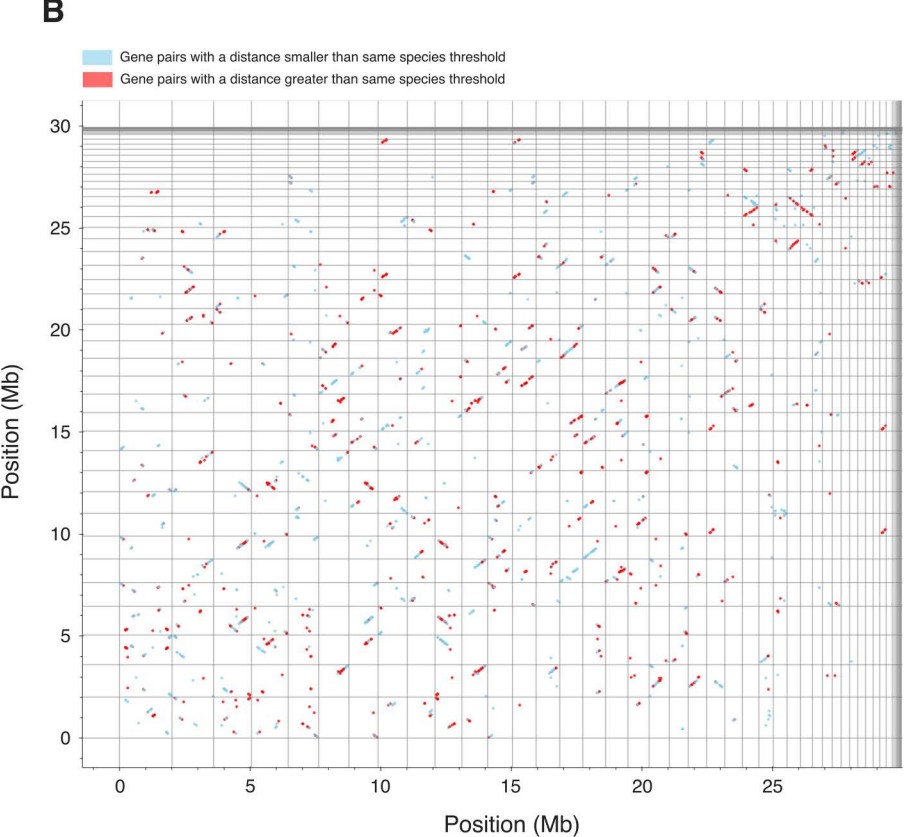

**Fig 3. Pairwise phylogenetic branch lengths between homeologous gene pairs in tetraploid genomes. (A)** Histograms showing phylogenetic branch lengths (in amino acid substitutions per site) between homeologous gene pairs for tetraploid genomes. The relaxed branch length threshold for species delineation is highlighted in a dashed red line (0.032 amino acid substitutions per site). The percentage of gene pairs that exceed this

same-species threshold is given in a box in the top right of each plot. **(B)** Oxford dot plot of tetraploid ilAceEphe1.μ (from host *Acentria ephemerella* [Lepidoptera]) using BUSCO genes. Contig boundaries are marked by gray lines. Gene pairs that are less divergent than the same species threshold are in sky blue, while gene pairs that are more divergent than the same species threshold are in red. The data underlying (A) was generated by running BUSCO (microsporidia_odb10, version 5.4.6) [76] on the unpurged genome assemblies of the tetraploid genomes. For each tetraploid, the haplotypes of each BUSCO locus were aligned to one another and an outgroup using MAFFT (version 7.525) [90], and a phylogeny was generated for each alignment using IQ-TREE (version 2.3.4, with ModelFinder enabled) [77, 91]. Subsequently, the branch lengths between homologous gene pairs were extracted from each phylogeny, and plotted in the histograms seen in (A) using a custom script in S1 Script. The individual BUSCO phylogenies used to derive this data can also be found in File Collection 15 at https://doi.org/10.5281/zenodo.17251512. The BUSCO gene annotations used to generate (B) can be found in File Collection 5 at https://doi.org/10.5281/zenodo.17251512. The figure was generated using Matplotlib [92], and manually annotated using InkScape (version 1.2.2).

We performed the Approximately Unbiased statistical test [94] on multi-copy BUSCO gene phylogenies for all pairwise combinations of tetraploid microsporidian genomes. We found that the haplotypes in each tetraploid were more similar to each other than they were to genomes from other species (i.e., haplotypes coalesce more recently than species) (Fig 4). In three species where we had multiple high-contiguity assemblies, the homeologous genomes had coalescences deeper than the within-species coalescence of homologs, implying a single origin of tetraploidy at the base of the species. Thus, the three *Vairimorpha cerenae* genomes, the two genomes assessed in gmOTU1 and the four genomes assessed in gmOTU2 had tetraploid origin coalescences deeper than the within-species coalescence (Fig 4). Interestingly, the three *Anncaliia algerae* genomes appear more distinct, suggesting reduced flow between the sampled individuals (Fig 4). As noted above, ilAceEphe1.μ may have arisen from a hybridization event (Fig 3). However, its diploid ancestors were likely more closely related to each other than they were to other tetraploid genomes sampled in our study, as we observed nearly all coalesces between the homeologues within the tetraploid occurring more recently than they do with other genomes (Fig 4).

Because all tetraploid microsporidian genome homeologous subgenomes coalesce more recently than species, we cannot distinguish between ancient tetraploidy in a system where recombination, gene conversion, and/or sexual reproduction homogenize the two subgenomes; and recent tetraploidy in a system that may or may not undergo recombination and/or sexual reproduction.

## Microsporidian genomes may carry signals of recent recombination

While our coalescence analyses above (Fig 4) and previous studies suggest that Microsporidia may undergo sexual reproduction [21,24,29–40,42–45], unequivocal signals of recombination have yet to be observed genomically. Phased chromosome-level genomes with known ploidies can help address this by identifying runs of homozygosity between otherwise differentiated homologous chromosomes that may have arisen by recent sexual or non-sexual (i.e., gene conversion) recombination.

For a purged tetraploid genome where all four copies of the genome were reconstructed (iuLoeVari1.μ from host *Loensia variegata* [Psocodea]), we assessed the nucleotide identity patterns between the homologous copies of the largest chromosome. It is striking that this analysis did not identify two pairs of diverged homeologues, but instead suggested a mosaic pattern of pairwise similarity (Fig 5). In line with this, we found nearly 20% of the tetraploid genome collapsed in the genome assembly (estimated haploid size is ~17 Mb, whereas tetraploid assembly span is 54.7 Mb). The collapsed regions identified are at the ends of chromosomes, where recombination rates are higher in many organisms [100,101]. Similar patterns were also observed in other high-contiguity genomes (File Collections 6 and 7 at https://doi.org/10.5281/zenodo.17251512). These observations are consistent with signatures of recent recombination.

## The tetraploid microsporidian cell contains two diploid units

Given the diversity of nuclear conditions in Microsporidia, we sought to understand 3D genome architecture in tetraploids, to determine whether homeologous subgenomes reside in distinct units (such as the nuclei of diplokaryons). Thus, we

PLOS Biology

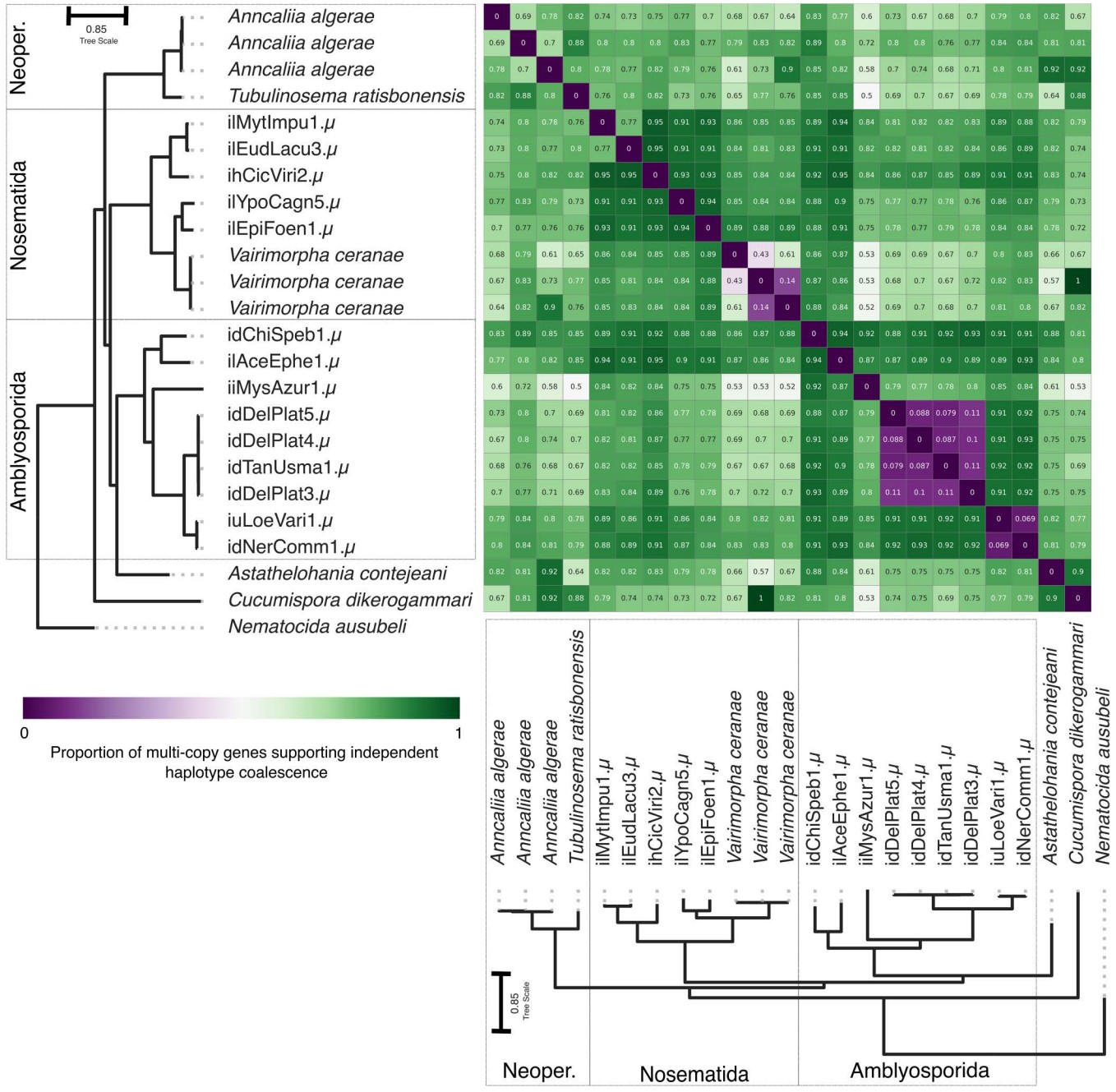

**Fig 4. Proportion of multi-copy genes which coalesce prior to genomes.** Heatmap showing the fraction of genes that support a more recent homeologue coalescence than between-species coalescence. Fractions greater than 50% are indicated in green, whereas fractions lower than 50% are indicated in purple. The phylogeny is an ASTRAL [75] phylogeny summarizing individual phylogenies of 600 BUSCO genes (microsporidia_odb10) [76] from all publicly available tetraploid assemblies and the tetraploid assemblies generated in this study. The branch lengths were estimated using a concatenated alignment of the individual BUSCOs used, with IQ-TREE [77]. The phylogeny is congruent with the phylogeny in Fig 2. The data underlying this figure can be found in S2 Text. The figure was generated using Matplotlib [92] and ToyTree [73], and manually annotated using InkScape (version 1.2.2).

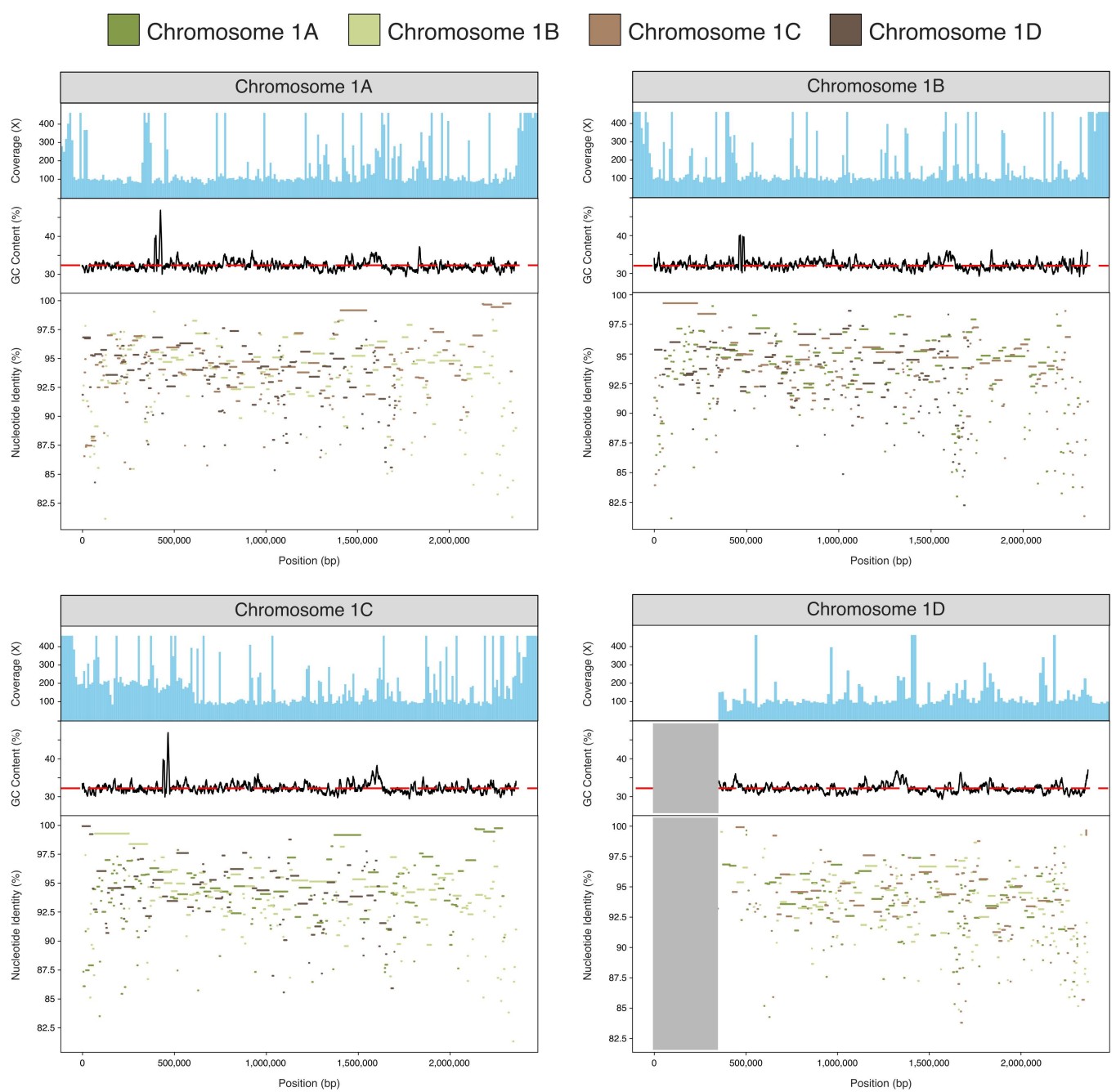

**Fig 5. Nucleotide identity (%) between tetraploid iuLoeVari1. μ haplotypes.** Each haplotype was compared to the other three haplotypes using minimap2 [95], and nucleotide identity (%) between them was plotted for each reference. Each haplotype has a mosaic pattern of identity to the others. The gray shaded area represents a "missing" segment of chromosome 1D, which we suggest is identical to and thus coassembled as the corresponding portion of chromosome 1C, which has double the expected coverage. The top panel of each plot shows mapped read coverage, and the middle panel displays GC content along the chromosome, with average GC content marked by a dashed red line. The coverage data underlying this figure was generated by mapping the PacBio reads against the genome using minimap2, and extracting read depth data using samtools and bedtools [96,97]. The GC data was generated by running seqkit fx2tab [98] on the genome. The genome can be found in File Collection 12 at https://doi.org/10.5281/zenodo.17251512. The genome's BioSpecimenID can be found in S1 Table, and can be used to retrieve the associated PacBio reads from NCBI [99]. The figure was generated using Matplotlib [92], and manually annotated using InkScape (version 1.2.2).

leveraged the availability of high coverage Hi-C data for one of our tetraploid microsporidian genomes, iuLoeVari1.μ from host *Loensia variegata* [Psocodea], to assess physical interactions (chromatin proximities) between the four subgenomes inside the microsporidian cell.

The Hi-C data indicated that for each chromosome, the four homeologous copies have a "two plus two" association (Fig 6). In addition, looking between chromosomes, these pairs are in turn more likely to interact with other pairs, and the whole genome can be partitioned into two diploid units, each containing 20 chromosomes (Fig 6). Furthermore, the signals consistent with recombination identified above (Fig 5) occur between the homeologous copies of chromosome 1 within and between these units. Taken together, our results suggest that the microsporidian tetraploid genome occurs in two recombining diploid units.

## No evidence of recent rediploidisation in Microsporidia

Given we do not see deep coalescence of homeologues in tetraploid microsporidians, and cannot distinguish ancient from recent tetraploidisation events in the group, we sought to test whether diploid lineages showed any signal of an ancient tetraploid state. If tetraploidy was ancestral to Microsporidia or to major lineages within the group, the diploid lineages derived from tetraploid ancestry should show evidence of rediploidisation. Rediploidisation could be achieved by reestablishment of a diploid karyotype by selective loss of one set of chromosomes or through meiotic reduction division without subsequent fertilization. Alternatively, diploidy could be restored piecemeal by loss or subfunctionalisation of the homeologous copies of each gene [104–108]. Piecemeal rediploidisation should leave a signal in the age of retained homeologous gene pairs, reflecting the divergence between the parental genomes, which would appear as paralogues in the diploidised genome.

We explored the age distribution of likely paralogous BUSCO gene pairs (homologous genes originating from a gene duplication event, called using wgd [104]) on the diploid microsporidian genomes. Relative age was estimated as the synonymous divergence (Ks) between the gene pairs. Piecemeal rediploidisation should be evident as a peak of gene duplicates of the same age [104–108]. Restoration of diploidy through reductive division would not be expected to leave behind such a signal, as the remaining paralogous gene pairs would have had independent origins. Similarly, the absence of an ancient tetraploid state in diploid lineages would not be expected to leave behind such a signal. We found no peaks of shared divergence in the diploid genomes (Fig 7). Thus, this suggests that tetraploidy in Microsporidia may result either from recent, independent polyploidisation events; or from an ancestral tetraploid state followed by rediploidisation in diploid lineages through a process similar to that of reductive division in gametogenesis or by inheriting a single nucleus of a diplokaryon.

## Evidence for rearrangements between homeologues and between genomes in Microsporidia

In addition to segmental duplications, we observe that some tetraploid microsporidian genomes carry signals of between-homeologue rearrangements. Those include inversions, fusions, fissions, and translocations that generated uneven haploid genomes that differed in gene content. To quantify how pronounced this phenomenon is in our genomes, we used a greedy algorithm to bin each tetraploid genome into four subgenome bins. Briefly, the algorithm iterates through contigs from largest to smallest, and appends a contig to a haplotype if the duplication in that contig does not exceed a specified threshold (see Github: https://github.com/Amjad-Khalaf/gerbil for implementation).

idChiSpeb1.μ (host *Chironomus sp.* [Diptera]) is a perfect tetraploid, with no assembly collapse, and thus can be binned into four equal subgenomes for the vast majority of duplication thresholds tested (Fig 8). Similarly, we were able to partition the genomes of iuLoeVari1.μ (host *Loensia variegata* [Psocodea]) and idNerComm1.μ (host *Neria commutata* [Diptera]) recovering two incomplete subgenomes showing some genome collapse (Fig 8). On the other hand, for three other genomes including ilAceEphe1.μ (host *Acentria ephemerella* [Lepidoptera]), ilMytImpu1.μ (host *Mythimna impura* [Lepidoptera]), and ihCicViri2.μ (host *Cicadella viridis* [Hemiptera]) we were unable to recover a haplotype with completeness similar to that of the unbinned genome assembly without also retaining a higher amount of duplication than expected (Fig 8 and S6 Fig). In line with this, the self-alignment dot plots of those genomes show numerous rearrangements and

**A**

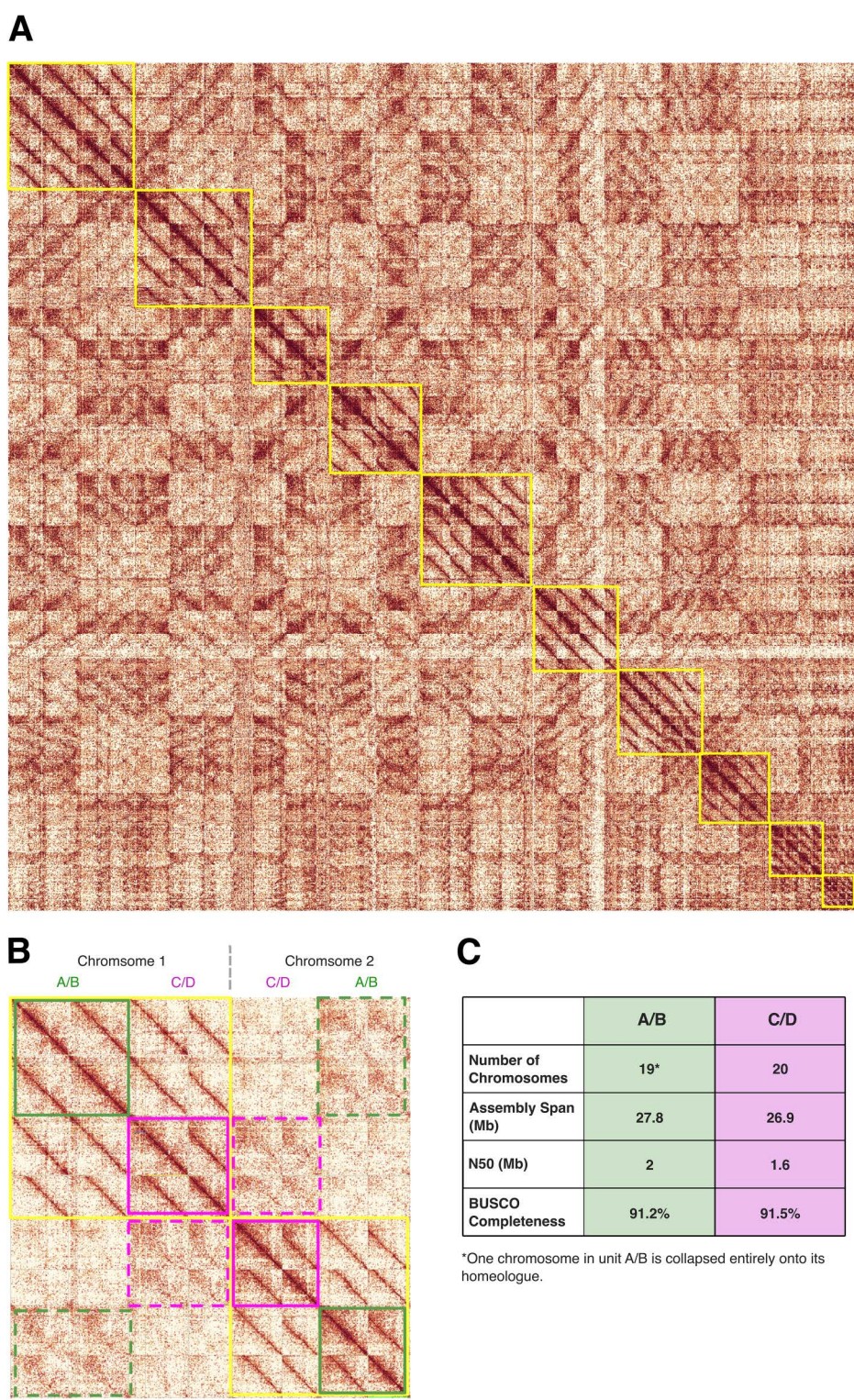

**B**

Chromsome 1     Chromsome 2
A/B        C/D        C/D        A/B

**C**

|  | A/B | C/D |
|---|---|---|
| **Number of Chromosomes** | 19* | 20 |
| **Assembly Span (Mb)** | 27.8 | 26.9 |
| **N50 (Mb)** | 2 | 1.6 |
| **BUSCO Completeness** | 91.2% | 91.5% |

*One chromosome in unit A/B is collapsed entirely onto its homeologue.

**Fig 6. Hi-C heatmap for the tetraploid genome of iuLoeVari1.μ.** Hi-C contact maps are heatmaps that visualize the frequency of physical contacts between genomic regions in 3D-space. Regions that are closer together physically tend to show more interactions, appearing as darker colors on the map. The strongest signal, in dark red here, is always found along the diagonal, which represents self–self interactions (i.e., each genomic region

interacting with itself and nearby regions along the same chromosome). Off-diagonal signals represent interactions between different chromosomes. **(A)** Hi-C contact map of the tetraploid iuLoeVari1.μ genome (host *Loensia variegata* [Psocodea]). Each chromosome, with its four copies, is highlighted by a yellow box. **(B)** Hi-C contact map showing the interactions amongst the four copies of chromosome 1 and the four copies of chromosome 2. Green lines highlight interactions belonging to unit 1, and purple lines highlight interactions belonging to unit 2. Dotted lines indicate interactions between chromosomes 1 and 2. **(C)** Summary metrics for the genome assemblies of units A/B and C/D. The data underlying this figure was generated by mapping the Hi-C reads to the genome using the sanger-tol/curationpretext pipeline [102] (excluding multi-mapping reads). The genome can be found in File Collection 12 at https://doi.org/10.5281/zenodo.17251512. The genome's BioSpecimenID can be found in S1 Table, and can be used to retrieve the associated Hi-C reads from NCBI [99]. The figure was generated using PretextView [103], and manually annotated using InkScape (version 1.2.2).

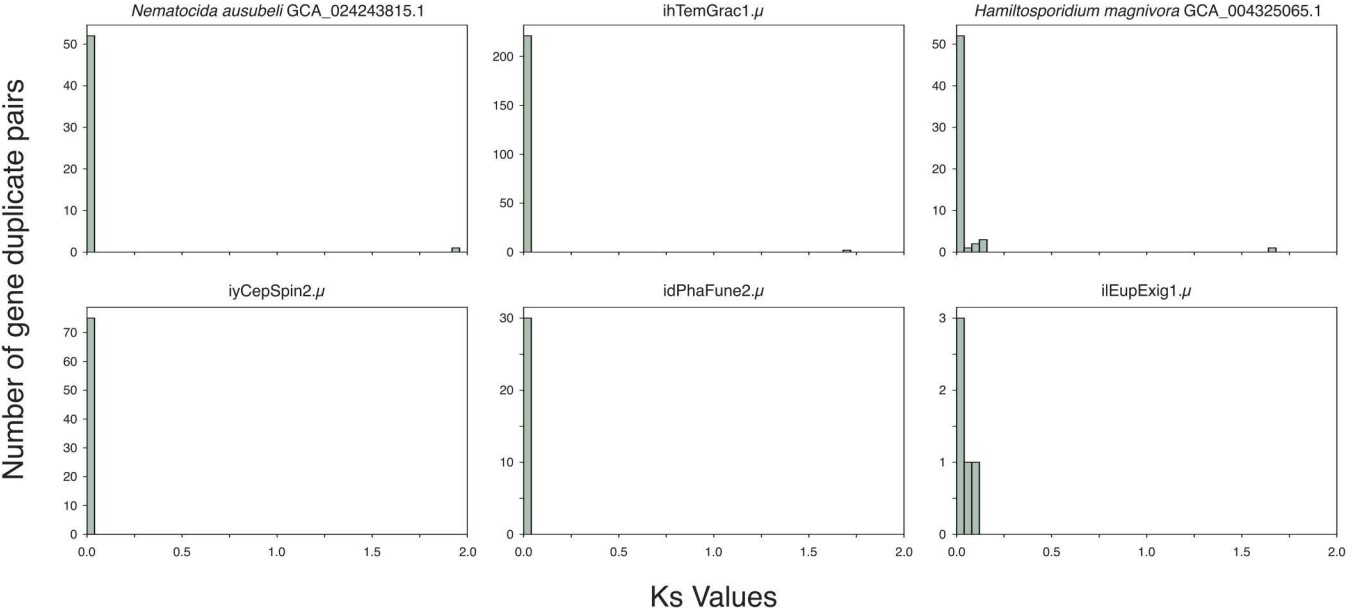

**Fig 7. Age distributions of duplicate gene pairs.** Histograms showing synonymous divergence (Ks) distributions for candidate paralogous BUSCO gene pairs from representative diploid genomes. No evidence of recent rediploidisation events is seen, as there are no peaks against a background exponentially-decaying distribution coming from small-scale gene duplication events. The y-axis is highly variable due to different BUSCO gene family expansions occurring in different lineages, yielding larger counts of possible paralogous gene pairs. wgd was used to identify paralogous genes in every genome and compute Ks values [104]. The data underlying this figure can be found in File Collection 16 at https://doi.org/10.5281/zenodo.17251512. The figure was generated using Matplotlib [92], and manually annotated using InkScape (version 1.2.2).

translocations (File Collection 6 at https://doi.org/10.5281/zenodo.17251512). We also found evidence of similar rearrangements and unevenness in other diploid and tetraploid purged microsporidian genomes, and in some genomes without known ploidies (Self alignment plots in File Collection 6 at https://doi.org/10.5281/zenodo.17251512). Together, these results provide evidence for extensive between-haploid subgenome rearrangement in microsporidian genomes, resulting in uneven gene distribution between haploid subgenomes. We examined whether between-haplotype rearrangements were associated with any genome metadata, and found no phylogenetic, host, or ecological association linking them.

Given the above observations of synteny breakage between subgenomes within tetraploids, we expect to observe fractured synteny between species. Thus, we mapped the relative position of conserved orthologous BUSCO genes in all chromosome-level genome assemblies. Comparing closely-related genomes showed a pattern of dynamic change of microsporidian linkage groups, and a high rearrangement rate throughout Microsporidia (Fig 9). Confident reconstruction of ancestral linkage groups for all Microsporidia was not possible, likely because the rearrangement rate was indeed too high (see S3 Text and S7–S12 Figs for details on rearrangements inferred and methods attempted).

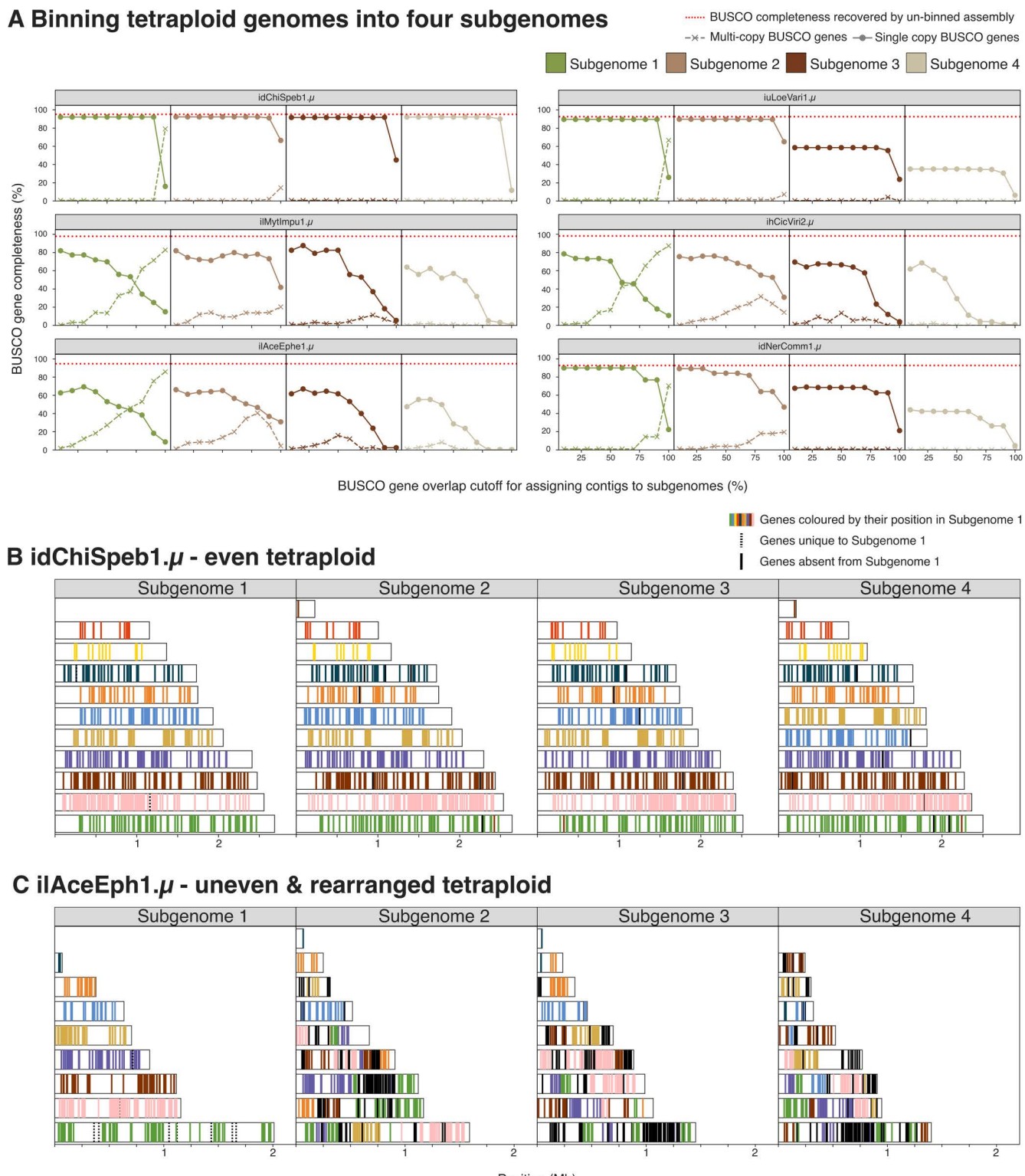

**Fig 8. Binning tetraploid genomes into four subgenomes using BUSCO genes. (A)** Using a greedy algorithm, we iterated through contigs from largest to smallest, appending a contig to a haplotypic subgenome if the duplication contributed by that contig does not exceed a specified threshold (*x* axes in the figure). Single-copy BUSCO gene completeness is marked by circles and multi-copy BUSCO gene completeness is marked by crosses.

A red dashed line denotes the BUSCO completeness score of the unbinned assembly. For **(B)** idChiSpeb1.μ and **(C)** ilAceEphe1.μ, we plotted the largest 10 contigs in subgenome 1 with their BUSCO genes, and coloured these genes in the other subgenomes by their positions in subgenome 1. The BUSCO annotations underlying this figure can be found in File Collection 5 at https://doi.org/10.5281/zenodo.17251512. The figure was generated using gerbil (https://github.com/Amjad-Khalaf/gerbil), and manually annotated using InkScape (version 1.2.2).

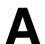
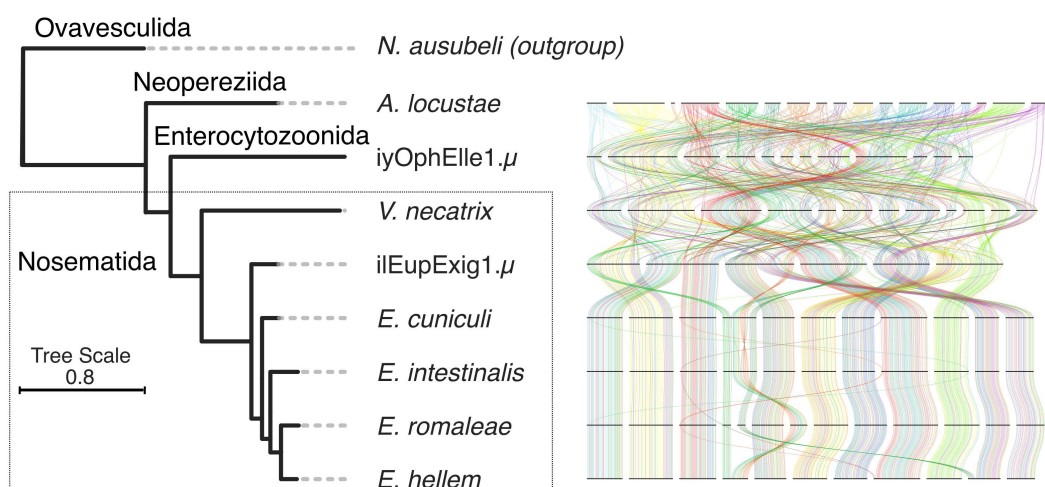
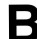
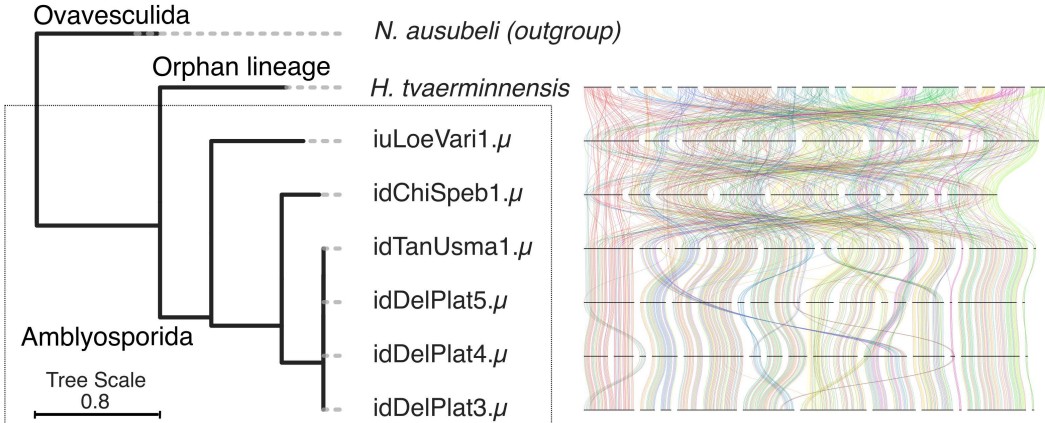

**Fig 9. Synteny plots of chromosomal microsporidian genome assemblies.** Genome-wide synteny plots of all chromosomal microsporidian genome assemblies for **(A)** Enterocytozoonida, Nosematida, and Neopereziida; and **(B)** Amblyosporida and the Orphan lineage. Each line represents a single-copy BUSCO (microsporidia_odb10) [76]. In (A) BUSCOs are painted by their chromosomal position in *A. locustae*, while in (B) they are painted by their chromosomal position in *H. tvaerminnensis*. The attached phylogeny is an ASTRAL phylogeny summarizing individual phylogenies of 600 BUSCO genes (microsporidia_odb10) [76]. The branch lengths were subsequently estimated using a concatenated alignment of the individual BUSCOs used, with IQ-TREE [77]. The BUSCO annotations underlying this figure can be found in File Collection 5 at https://doi.org/10.5281/zenodo.17251512. The figure was generated using the ribbon plotting script in https://github.com/conchoecia/odp [109] and ToyTree [73], and manually annotated using InkScape (version 1.2.2).

## Discussion

### Forty new genome assemblies, a revised phylogeny, and the future of species delineation in Microsporidia

In this study, we present 32 new complete microsporidian genome sequences (including eight chromosome-level genomes, seven of which, to our knowledge, were scaffolded with the first implementation of Hi-C data on Microsporidia), and eight additional partial genome sequences (with BUSCO completeness score <70%). While not being the target organisms in the sequencing efforts used to generate these genome assemblies, most of the assemblies presented in this study have comparable or higher contiguity and BUSCO completeness than published microsporidian genomes (Fig 2). Our genome assembly process provides an accessible and successful approach for the assembly of microsporidian genomes from their host sequencing data, and can be translated to other cobionts where a wealth of sequenced host genomes is available.

The presented microsporidian genomes derive from five of the seven major microsporidian clades as defined by Bojko and colleagues [4]. They allowed us to revise microsporidian phylogeny, notably confirming Neopereziida as a sister group to the ancestor of Nosematida and Enterocytozoonida (Fig 2).

We show that genomic data support previous species delineations based on morphology, histopathology, and cell culture, and suggest divergence thresholds that differentiate species. While morphological and histopathological data remain important for identifying Microsporidia to species, it is likely that serendipitous discovery through large-scale genomic sequencing will be a major source of new microsporidian isolate genomes in the future. In prokaryotes, genomic metrics for species delineation, such as Average Nucleotide Identity (ANI) [110], are well accepted. ANI has also been used previously on Microsporidia [111], and the application of similar metrics to taxon delimitation in eukaryotes is in its infancy [112].

### Tetraploidy in Microsporidia

From theoretical and observational grounds, the long-term maintenance of tetraploidy is unlikely. Usually, a new tetraploid lineage will revert to effective diploidy, either by losing one set of chromosomes or by reestablishment of exclusive pairing between homologs and doubling of the diploid chromosome number. The reestablishment of diploidy (rediploidisation) leaves a signature of whole genome duplication (WGD) in descendants, involving loss of copies of many genes and subfunctionalisation of retained genes [113]. WGD has been common during evolution of plants, animals, and fungi, and has been proposed to be a significant driver of organismal complexity and adaptive evolution [56]. The situation in Microsporidia differs from classical yeast, flowering plant, and animal models of tetraploidy and rediploidisation, as the constituent genomes appear to be largely maintained intact and show signatures of ongoing recombination between all four genomes.

In some microsporidian genomes, we observed nearly perfect synteny between haplotypes. However, we observed a striking level of between-haplotype rearrangement in three microsporidian genomes, two autotetraploids (ilMytImpu1.μ and ihCicViri2.μ) (Fig 8), and a genome with circumstantial evidence of being the result of a recent hybridization event between two closely related diploid genomes (Figs 3 and 8). We also see similar signatures, albeit on smaller scales, in other genomes (Self alignment plots in File Collection 6 at https://doi.org/10.5281/zenodo.17251512). In other taxa, such rearrangement is often associated with hybrid origin in polyploids, for example, in synthetic plant tetraploids such as *Brassica* allotetraploids [114–118] and autotetraploid *Arabidopsis thaliana* [119], and in the multiple origins of tetraploidy in lager brewing yeast *Saccharomyces pastorianus* [120–122]. Such rearrangement may be a common genomic response to the "genome shock" of the origin of tetraploidy [123,124].

We cannot currently distinguish a general genomic shock hypothesis for the origin of these rearranged microsporidian genomes from other possibilities, such as independent genetic disruption of reproductive processes, as seen in a number of yeast isolates with similar karyotype patterns [125]. Population-level whole-genome sequencing of selected species, over time (both in nature and in lab cell cultures), is required to better define this phenomenon and identify its root causes.

## A diploid/tetraploid mating system as a potential reproductive model for Microsporidia

Understanding the origin of tetraploidy and its biology in Microsporidia has significant implications. If tetraploidy is ancient, Microsporidia would represent a rare case of a stable, species-rich tetraploid lineage, offering a unique opportunity to study genome evolution under these conditions. On the other hand, if tetraploidy is recent, Microsporidia would represent a clade with an unusually high propensity for polyploidisation. Additionally, if ploidy influences host specificity, this could inform strategies to mitigate the impact of Microsporidia on aquaculture and beekeeping or their potential as biological control agents, such as for malaria. Furthermore, a deeper understanding of their reproductive behavior could also aid in both managing and leveraging these organisms.

We identified many tetraploid microsporidian genomes, the majority of which are likely autotetraploids (Fig 3), but no concrete evidence that tetraploidy was the ancestral state in Microsporidia. We observed that homeologous subgenomes coalesce more recently than do the genomes of different species in all cases (Fig 4). However, this does not allow us to distinguish between ancient shared tetraploidy versus recent independent tetraploidy, as both models are likely to be indistinguishable in the presence of recombination within and between homeologous subgenomes during sexual reproduction. Similarly, in diploid lineages nested within tetraploids, we did not observe signals congruent with classic piecemeal rediploidisation (Fig 7). This suggests that tetraploidy in Microsporidia is independently acquired in the polyploid clades, or that diploid lineages have restored diploidy through one-step mechanisms that eliminate one diploid set of chromosomes. This could arise if a lineage was founded from one diploid unit (Fig 6), or through meiotic reduction division without subsequent fertilization.

If tetraploidy is the result of recent, independent events, a minimum of 15 events is predicted. We thus suggest that it is more likely that a propensity for tetraploidy is ancient in Microsporidia, and that the diploid lineages nested within tetraploids represent isolates or lineages that have undergone reductive division.

One of our key findings is that tetraploid microsporidian genomes are likely organized into two diploid units (Fig 6), with evidence of recombination both within and between units (Fig 5). Taking all our findings together, we propose that the units are the two nuclei of a diplokaryon (i.e., each nucleus is diploid), that Microsporidia undergo occasional sexual reproduction in a process that mirrors fungal reproduction (Fig 10), and that chromosomes independently reassort into the units in different individuals. Our interpretations of the data are in line with previous life cycle proposals of the diplokaryon being tetraploid, and a diploid/tetraploid cycling underpinning microsporidian life [51,52]. Furthermore, there is extensive morphological evidence of plasmogamy (cell fusion) and karyogamy (nuclear fusion) in Microsporidia [21,24,29–40,42,43], and these processes may reflect this proposed sexual cycle. We note that there is some Hi-C signal between sequences belonging to different diploid units in Fig 6. We propose that this is the result of read mismapping, or compromised membranes between the two proposed nuclei. The latter may occur as a result of the fixation process in some of the cells, or as a genuine biological phenomenon. For instance, gaps between nuclei membranes have been noticed in two diplokarya [29,43]. Similarly, there is also evidence of Hi-C signal across nuclei in dikaryotic fungi [126].

In this model, both diploids and tetraploids can function as infective agents (Fig 10) with mitotic capabilities, aligning with the previously proposed period of rapid clonal propagation by Corradi (2015) [52]. Support for this proposal also comes from life cycle observations. Several microsporidians exhibit complex life cycles, generating multiple spore types with different infective potentials from a variety of hosts. For example, diplokaryotic *Edhazardia aedis* spores infect adult mosquitoes, producing monokaryotic spores that go on to infect larvae, generating diplokaryotic spores [31]. *Amblyospora connecticus* cycles between mosquito and copepod hosts using monokaryotic and diplokaryotic spores for infecting each host, respectively [127]. This alternation of generations with different ploidies could explain why the four genomes in tetraploids remain intact, and have not been subjected to the usual processes of rediploidisation, as random assortment of genomes into the diploid would select for fully functional diploids whichever pair was combined.

In the proposed model, each diplokaryon nucleus is diploid, and their mitosis requires only pairing of two homologous chromosomes, as in any diploid. In the proposed tetraploid fusion, however, pairing and correct partitioning of the

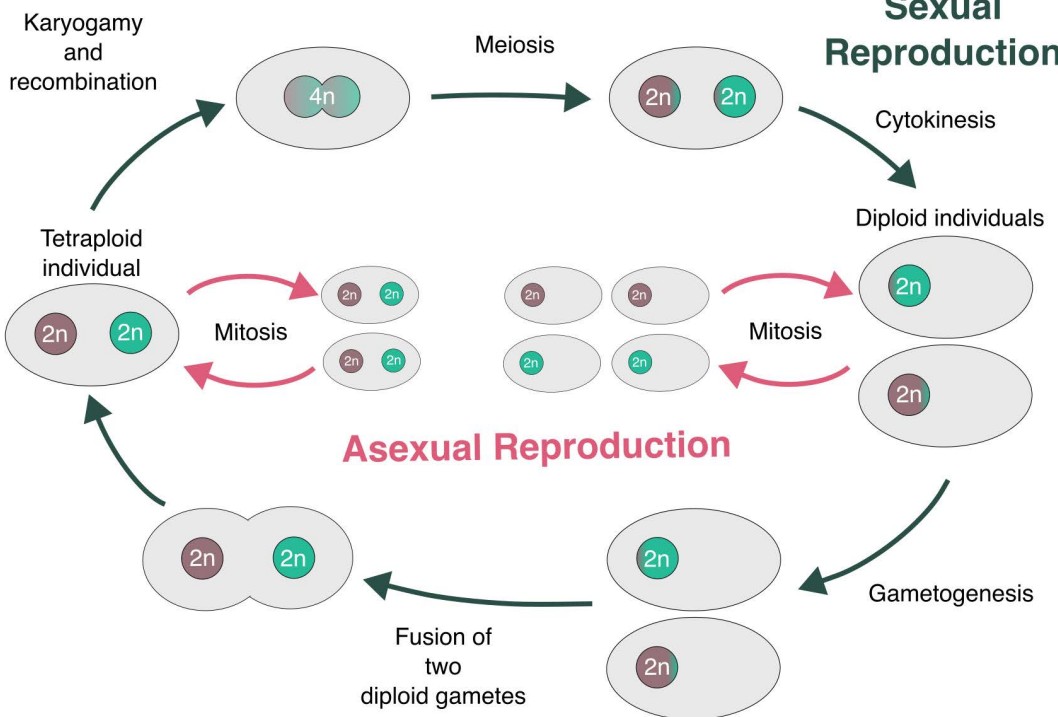

**Fig 10. Simplified proposed generalized lifecycle for Microsporidia.** Our proposed model posits that each nucleus is a diploid, and that microsporidian reproduction mirrors reproduction in Fungi with stages similar to karyogamy, plasmogamy, and a stable "heterokaryon" (known as a diplokaryon in Microsporidia). Importantly, both the diplokaryotic and monokaryotic phases are parasitic, and species may spend most of their lifecycle in one or the other phase, giving rise to "diploid" and "tetraploid" lineages. The figure was manually drawn using InkScape (version 1.2.2).

chromosomes to diploid daughter nuclei requires that all four homologous chromosomes are recognized and sorted coherently. In diploid cells, homologous chromosome pairing is mediated through sequence recognition. Minimally, the four copies of each chromosome in the proposed tetraploid fusion must carry some signature that means they can be recognized even if they diverge in sequence elsewhere such that they can no longer recombine.

Our model predicts that there should be diploid, "gametic" forms for all the tetraploid Microsporidia, and that tetraploid forms may exist for the described diploid lineages. We note that it remains unclear how ploidy maps on to life stages in most species, and this information is crucial to underpin the nature of microsporidian reproduction. Generation of single-cell, whole-genome data for Microsporidia, focusing especially on species with complex life cycles and multiple spore types would be highly informative.

While the diploidy of the microsporidian nucleus is broadly in line with morphological data, with most diploid species being monokaryotic, and most tetraploid species being diplokaryotic [53,54], there are two exceptions. Namely, tetraploid *Agmasoma penaei,* which possesses monokaryotic spores (suggesting each nucleus is tetraploid), and diploid *Vittaforma corneae,* which possesses diplokaryotic spores (suggesting each nucleus is haploid) [53,128]. Given the morphological descriptions are not from the same isolates used to predict their ploidy, it is possible that these species are polymorphic, as observed in Microsporidia with complex, alternating life cycles and multiple spore types [41,129]. Future sampling of those lineages in particular will be crucial to whether each nucleus is truly diploid in Microsporidia.

In this work, we generated 40 high-quality genome assemblies, found signals consistent with recombination, and provided evidence suggesting that microsporidians are likely autotetraploids. While the timing and number of polyploidisation

events remains uncertain, we propose that tetraploidy is an ancient feature of Microsporidia, with diploid lineages representing "reduced" forms. Population-level whole-genome sequencing, combined with longitudinal imaging in nature and laboratory cultures, will be crucial to illuminate the true nature of polyploidy, and its relationship to the lifecycle in Microsporidia.

## Materials and methods

### Data

Samples sequenced as part of the DToL project [68] were processed by the Tree of Life core laboratory and the Scientific Operations core at the Wellcome Sanger Institute. This process typically relies on using different parts of one individual, or two closely related individuals, to generate long-read DNA sequencing data and Hi-C short-read sequencing data. Because of this, specimens identified as microsporidian-infected from long-read sequencing data did not often have Hi-C data available.

To ensure we were able to generate Hi-C data for a subset of the microsporidian genomes we explored, we sampled 650 individual flying insects on the Wellcome Genome Campus using a Malaise trap in the summer of 2023. These specimens were first bisected, and DNA for long-read sequencing was extracted from one half. The other half was kept at −80°C for later Hi-C sequencing. We identified infected insects by PCR amplification testing of the long-read DNA extracts using a microsporidia-specific amplicon locus targeting the small subunit (SSU) rRNA hypervariable V1-V3 regions [130,131]. The primers used were V1F and R30R, with standard protocol as outlined in the literature [130,131]. Six specimens were identified as microsporidia-positive, and library preparation and genome sequencing of those six individuals, with ToLIDs idDelPlat3.µ, idTanUsma1.µ, idChiSpeb1.µ, idDelPlat4.µ, idLucSpea1.µ, and idDelPlat5.µ, was performed by the Scientific Operations core at the Wellcome Sanger Institute.

BioSpecimen identifiers for each dataset used in this study are listed in the S1 Table, along with the microsporidian genome assemblies.

### ToLIDs and OTUs

In the manuscript, we use Tree of Life identifiers (ToLIDs) of the host individuals that were sequenced at the Wellcome Sanger Institute, with the suffix ".µ", to refer to the microsporidian genome assemblies that resulted from them (see https://id.tol.sanger.ac.uk/ for more information). The full IDs the microsporidian genome assemblies are released under are listed in S1 Table, and do not contain the suffix ".µ". Additionally, we use "gm" as a prefix for each OTU, in line with ToLID notation (g for "Fungi", and m for "Microsporidia").

### Genome assembly

We identified 34 individual specimens sequenced for DToL as likely to be infected with a microsporidian using a MarkerScan [72] screen of their preliminary genome assemblies (generated using hifiasm version 0.19.9 [132]). DToL genome assemblies were screened on a rolling basis as they were generated, with a total of 1,200 genome assemblies screened to identify the infected specimens. For each infected individual, we concatenated the primary and alternate preliminary genome assemblies.

To positively identify microsporidian sequences, BlobToolKit [133] was run on each concatenated preliminary genome assembly. Contigs were filtered ("Filtering Step 1" in S2 Table) by a combination of average read coverage, GC content, and taxonomic classification of contigs using upper and lower bounds that retained contigs that had BLASTx matches mostly to microsporidian proteins, and excluded contigs that had a majority of database matches to proteins from to other taxa. The specific filtering parameters used for each genome assembly are reported in S2 Table.

The PacBio single-molecule HiFi long reads belonging to the preliminary assembly were aligned to the filtered microsporidian contigs using minimap2 (version 2.28) [95], and aligned reads were isolated using samtools (version

1.19.2, MAPQ = 255) [97]. To assess read coverage and ploidy, a k-mer spectrum was generated for the isolated reads using Jellyfish ($k$ = 21, version 2.2.10) [134] and analyzed using GenomeScope2 (version 2.0) [82] (File Collection 1 at https://doi.org/10.5281/zenodo.17251512). Where samples had high average read coverage (>20×) and reliable ploidy estimation, Smudgeplot (version 0.4.0 "Arched") was also run to confirm the ploidy assessment [82] (File Collection 8 at https://doi.org/10.5281/zenodo.17251512). We note that GenomeScope2 and Smudgeplot rely on heterozygosity for ploidy estimation [82], and while individuals with low heterozygosity can still have their ploidy correctly estimated, individuals with exactly 0% heterozygosity, such as some haploid selfing eukaryotes, are likely to be mis-classified.

The isolated PacBio HiFi reads were then reassembled using hifiasm (version 0.19.9, parameter −l0 was used to disable purging) [132]. The contigs from the reassembly were filtered again (using https://github.com/Amjad-Khalaf/BubblePlot, "Filtering Step 2" in S2 Table) by average read depth, GC content, and taxonomic classification of contigs, with upper and lower bounds that retained all contigs that contained microsporidian BUSCO proteins (microsporidia_odb10, version 5.4.6) [76], and excluded contigs that contained proteins mostly belonging to other taxa. The specific filtering parameters used for each genome assembly are reported in S2 Table. Assembly quality was subsequently evaluated using MerquryFK (Github: https://github.com/thegenemyers/MERQURY.FK) (File Collection 9 at https://doi.org/10.5281/zenodo.17251512).

In cases where ploidy was successfully estimated from the isolated reads, the ploidy status of the cleaned reassembly was assigned, or marked as unresolved ("NA" in S1 Table).

If Hi-C data was available for the same individual that the long read data had been generated from, Hi-C reads were mapped to the cleaned reassembly using bwa-mem2 (version 2.2.1) [135], and the resulting alignment files were converted to contact maps using Juicer Tools (version 1.8.9) [136], bedtools (version 2.31.1) [96], and PretextMap (version 0.19) [137] (using the sanger-tol/curationpretext pipeline [102]). If Hi-C data provided sufficient signal, contigs were then manually scaffolded using PretextView (version 0.2.5) [103] (excluding multi-mapping Hi-C reads). A haploid representation of the genome assembly was generated by selecting the most contiguous copy of each chromosome with the least gaps. This was possible for seven microsporidian genome assemblies.

If Hi-C data was not available, but the cleaned reassembly possessed sufficient coverage to estimate ploidy using GenomeScope2 (version 2.0) and Smudgeplot (version 0.4.0 "Arched") [82], purge_dups was used to generate a haploid representation of the genome assembly [80] (File Collection 10 at https://doi.org/10.5281/zenodo.17251512). This process was followed for eight microsporidian genome assemblies.

In the cases where Hi-C data was not available, or the cleaned reassembly did not possess sufficient coverage to estimate ploidy, or the cleaned reassembly showed high levels of rearrangements between its subgenomes (regardless of having been assigned a ploidy), purge_dups [80] was not run. For these assemblies, the unresolved cleaned reassembly was considered the final assembly and used in further analyses. This was the case for 27 microsporidian genome assemblies. The data available and the paths followed for each microsporidian genome assembly are outlined in S1 Table.

The genome assembly and read metrics for all intermediate steps for each microsporidian genome assembly are reported in File Collection 11 at https://doi.org/10.5281/zenodo.17251512. The genomes generated in this study are available in File Collection 12 at https://doi.org/10.5281/zenodo.17251512.

## Confirming that microsporidian genomes are derived from single species

We employed several checks to implicitly confirm that our microsporidian genomes are derived from samples with single-species infections. Firstly, the MarkerScan screen relies on identifying microsporidian ribosomal sequences in the arthropod reads, and would have identified multiple species of microsporidia had they been present in the sample [72]. Secondly, BlobToolKit plots would have highlighted whether identified microsporidian sequences belonged to different species. This is because any two co-infecting organisms are unlikely to maintain identical loads in their hosts, and they are unlikely to have identical sequence composition. Similarly, a mixed infection (whether it be multiple species or multiple

strains of the same species) would have been evident in the GenomeScope2 histograms. We have explored this in detail previously (see [53] for examples of mixed infections). Furthermore, in the case of polyploid genomes, our gene-based species-delineation analyses would have highlighted the presence of multiple species. Thus, we determined that all of our 40 microsporidian genomes are derived from single-species infections. However, we note that it is not possible to distinguish whether genomes with low coverage came from single, clonal infections or infections involving multiple, closely-related strains of the same species.

## Publicly available microsporidian genome assemblies

On the 1st of January 2025, we downloaded all microsporidian genome assemblies available in the NCBI Genome database. This retrieved 106 genome assemblies, whose accession numbers are listed in S5 Table.

## Alignment, phylogeny, and species delineation

Unless stated otherwise, all phylogenies were generated by identifying orthologous proteins using BUSCO (microsporidia_odb10, version 5.4.6.) [76], aligning each locus across all genomes using MAFFT (version 7.525) [90], inferring a tree for each of them using IQ-TREE (version 2.3.4) [77], and summarizing all the resulting gene trees into one species tree using ASTRAL (version 5.7.8) [75]. The branch lengths were subsequently estimated using a concatenated alignment of the individual BUSCOs used, with IQ-TREE [77]. The model chosen according to IQ-TREE's model finder was "Q.yeast.I.G4". In the case of multi-copy genes, one of the copies was chosen randomly. This was done because the majority of multi-copy genes across all the genomes showed that haplotypes displayed clear monophyly (see Results).

To infer species membership of unidentified genomes, we calculated average pairwise divergence between all possible genome pairs using the BUSCO gene phylogeny, and defined upper bounds for within-species divergence based on the genomes from identified microsporidian species from public data. We also explored generating a same-species threshold for each gene, using the distribution of branch lengths between genomes classified as belonging to the same species. Our results were consistent with those based on the whole-genome phylogeny branch lengths (see S4 and S5 Figs).

Other genomic approaches for species delineation, such as ANI, have been applied to Microsporidia [111]. However, we have elected to use amino acid substitutions per site on a BUSCO gene phylogeny because it relies on pre-established orthology, rather than the assumption that reciprocal blast hits represent orthologous sequences. Furthermore, many ANI approaches do not take into account the proportion of the genome that is represented in the reciprocal blast hits which are assumed to be orthologous. Thus, relying on amino acid substitutions per site in a universal set of orthologous genes like BUSCOs is less susceptible to noise.

## Assessing if haplotypes coalesce prior to genomes

For every possible combination of any two tetraploid genomes from the list of complete genomes generated in this study, and all publicly available genomes, a tree topology test was performed using IQ-TREE (version 2.3.4) [77] on multi-copy BUSCO gene trees. Support for the topology where each genome's haplotypes displayed clear monophyly was statistically assessed using the Approximately Unbiased Test [77,94]. Unpurged genome assemblies were used where available.

## Inference of historical rediploidisation events

To infer potential historical de-polyploidisation events, wgd (version 2) was run on each genome from the list of complete genomes generated in this study, and all publicly available genomes, to infer paralogous genes and compute their Ks values [104].

## Genome annotation

Each final microsporidian genome assembly was annotated for repeats using RepeatModeler (version 2.0.5) and Repeat-Masker (version 4.1.7) [78,79]. Repeat annotations for each genome assembly are reported in File Collection 13 at https://doi.org/10.5281/zenodo.17251512. We also annotated each final genome assembly for coding sequences using BRAKER2 [138], with protein hints from UniProt [139]. The resultant GeneMark-ES [140] annotation files can be found in File Collection 14 at https://doi.org/10.5281/zenodo.17251512.

## Between-haplotype rearrangements

To illustrate between-haplotype rearrangements in some of our genomes, we developed a greedy algorithm that bins each of our unpurged tetraploid genomes into four subgenomes based on BUSCO gene completion and duplication (see Github: https://github.com/Amjad-Khalaf/gerbil for implementation). After sorting contigs by from largest to smallest, our algorithm iterates through contigs and assigns a contig to a subgenome bin if the duplication that contig would add to its proposed subgenome bin does not exceed a specified threshold.

The discussed patterns can be highlighted by examining synteny between the recovered haplotypes for one of the duplication thresholds tested (the x-axis in Fig 8). For idChiSpeb1.µ and ilAceEphe1.µ, we plotted the largest 10 contigs in haplotype 1 with their BUSCO genes, and coloured these genes in the other haplotypes by their positions in haplotype 1 (Fig 8). As expected, the haplotypes of idChiSpeb1.µ show nearly perfect synteny with one another. However, ilAceEphe1.µ display a large number of rearrangements and haplotype-unique BUSCO genes (as seen by the pattern in Fig 8). In line with this, nearly all of idChiSpeb1.µ's BUSCO genes are in 4 copies, distributed across 4 haplotypes (S6 Fig). On the other hand, the vast majority of ilAceEphe1.µ's BUSCO genes are in less than 4 copies (despite its haploid coverage exceeding 48X, as seen in File Collection 1 at https://doi.org/10.5281/zenodo.17251512), and its BUSCO genes are not evenly distributed across its haplotypes. For instance, some BUSCO genes occur in 3 copies, all present in a single haplotype (S6 Fig). Together, these observations highlight the degree of rearrangement and unevenness in ilAceEphe1.µ, and similar scenarios are seen for ilMytImpu1.µ and ihCicViri2.µ.

## Between-genome rearrangements

We mapped the relative position of conserved orthologous BUSCO genes in all chromosome-level genome assemblies, using ribbon plot scripts from https://github.com/conchoecia/odp [98], to compare synteny patterns in Microsporidia. To quantify and synthesize the observed patterns with an evolutionary perspective, we also ran syngraph to infer a set of putative ancestral linkage groups for these genomes [141]. Additionally, we ran an unsupervised clustering of loci based on their chromosomal occupancy [142,143], also aiming to highlight any recoverable putative ancestral linkage groups.

## Supporting information

**S1 Table. Microsporidian genome assembly statistics and host metadata.** Full list of recovered microsporidian genome assemblies, their associated metadata, and host metadata.
(XLSX)

**S2 Table. Filtering parameters used in generating genome assemblies.** Parameters used for filtering microsporidian contigs from their respective (meta-)genomic assemblies in filtering steps 1 (BlobToolKit [133]) and 2 (BubblePlot, Github: https://github.com/Amjad-Khalaf/BubblePlot). See Materials and methods for details.
(XLSX)

**S3 Table. Trait-phylogeny regression.** Transformations representing the fit with the tree's topology ($\lambda$), branch-lengths ($\kappa$), and root–tip distance ($\delta$) [83] and the number of coding sequences, transposable element loads, and genome spans.
(XLSX)

**S4 Table. Trait correlation.** Correlations between transposable element loads and genome spans. (XLSX)

**S5 Table. Accession numbers for publicly available genomes used in this study.** On the 1st of January 2025, we downloaded all microsporidian genome assemblies available in the NCBI Genome database. This retrieved the following 106 genome assemblies. (XLSX)

**S6 Table. Branch length distances for species delineation.** Pairwise branch length distances which include one of our genomes, and can be classified to a species or a genus. The conservative branch length threshold range was defined using the shortest observed branch lengths between known same-species genomes for the lower bound (0) and the smallest distance between *H. tvaerminnensis* and *H. magnivora* genomes for the upper bound (0.012). The relaxed threshold uses the full range of observed branch lengths among known same-species genomes (excluding the *H. tvaerminnensis* – *H. magnivora* cutoff). (XLSX)

**S1 Text. Newick string of phylogeny.** ASTRAL [75] phylogeny summarizing individual phylogenies of 600 BUSCO genes (microsporidia_odb10) [76] across all publicly available microsporidian genome assemblies (*n* = 106), and the genome assemblies generated in this study (*n* = 40, marked in purple). Branch lengths were estimated with IQ-TREE using a concatenated alignment of the individual BUSCOs [77]. The model chosen according to IQ-TREE's model finder was "Q.yeast.I.G4". (TXT)

**S2 Text. Approximately Unbiased phylogenetic test results.** BUSCO (microsporidia_odb10, version 5.4.6) (Simão and colleagues 2015) was run on the unpurged genome assemblies of the tetraploid genomes. The haplotypes of each BUSCO locus were aligned to one another using MAFFT (version 7.525) (Katoh and colleagues 2002), and a phylogeny was generated for each alignment using IQ-TREE (version 2.3.4, with ModelFinder enabled and 1,000 bootstrap replicates) (Minh and colleagues 2020; Kalyaanamoorthy and colleagues 2017). The Approximately Unbiased statistical test [90] on the multi-copy BUSCO gene phylogenies for all pairwise combinations of tetraploid microsporidian genomes. The high-level summary of these pairwise tests are included in this text. For each listed pairwise comparison, the "+" sign indicates the number of phylogenies where haplotypes coalesce more recently than species, and the "−" sign indicates the number of phylogenies where species coalesce more recently than haplotypes. (TXT)

**S3 Text. Details on rearrangements inferred and methods attempted.** (TXT)

**S1 Script. Plotting histograms depicting phylogenetic branch lengths (in amino acid substitutions per site) between homeologous gene pairs for 13 tetraploid genomes.** Python script used to extract pairwise branch lengths between homeologous gene pairs for 13 tetraploid genomes, and plot them as histograms. Please note that the following genomes are represented by deprecated ToLIDs, which differ from the ones used in this manuscript. iyOecSmar33: idDelPlat3; iyOecSmar35: idTanUsma1; iyOecSmar39: idChiSpeb1; iyOecSmar41: idDelPlat4; and iyOecSmar44: idDelPlat5. (PY)

**S1 Fig. Sex of hosts the microsporidian genome assemblies are derived from.** The sex of our genomes' hosts was unknown in most cases (24 species). In the remaining cases, nine were identified as female and seven as male. A relatively equal proportion of female and male hosts are infected with Nosematida (Fig 1), but we could not assess skews in host sex ratios for other microsporidian groups due to missing data on sex (for Amblyosporida-infected hosts), or a small

sample size (for Neopereziida-infected hosts). The data underlying this figure can be found in S1 Table. The figure was generated using Matplotlib [92], and manually annotated using InkScape (version 1.2.2).
(PNG)

**S2 Fig. Ploidy inference examples for three microsporidian genomes, highlighting segmental duplications.** GenomeScope2 transformed linear plot and Smudgeplot [82], respectively, for **(A)**, **(B)** diploid iyCepSpine2.μ (host *Cephus spinipes* [Hymenoptera]); **(C)**, **(D)** diploid idPhaFune2.μ (host *Phania funesta* [Diptera]); and **(E)**, **(F)** polyploid (tetraploid or octoploid) iiMysAzur1.μ (host *Mystacides azureus* [Trichoptera]). Jellyfish was used to generate the initial k-mer spectra ($k=21$, version 2.2.10) [134]. Both iyCepSpin2.μ and idPhaFune2.μ have mostly diploid genomes, but carry a level of duplication that generated an identifiable "tetraploid" signal in their k-mer spectra. Similarly, the k-mer spectrum of iiMysAzur1.μ can be interpreted as either a highly homozygous tetraploid where large segmental duplications have occurred in all the four copies leading to a detectable octoploid signal, or an octoploid genome composed of two distinct tetraploids. Such cases are common, with some level of segmental duplication observed in nearly all of the 14 polyploid genomes (refer to File Collection 1 at https://doi.org/10.5281/zenodo.17251512). The genomes used to generate this figure can be found in File Collection 12 at https://doi.org/10.5281/zenodo.17251512. The figure was generated using GenomeScope2 [82], and manually annotated using InkScape (version 1.2.2).
(PNG)

**S3 Fig. 600 Gene Phylogeny of Microsporidia. (A)** ASTRAL [75] phylogeny summarizing individual phylogenies of 600 BUSCO genes (microsporidia_odb10) [76] across all publicly available microsporidian genome assemblies (including multiple strains where they are available), and the genome assemblies generated in this study ($n=40$, marked in purple). Branch lengths were estimated with IQ-TREE using a concatenated alignment of the individual BUSCOs [77]. Nodes with less than 95% support are marked with pink circles. Ploidy is marked in circles at the tips of the tree for genomes where it was characterizable. **(B)** Genome assembly span (Mb) as calculated by assembly-stats (Github: https://github.com/sanger-pathogens/assembly-stats), with black circles marking chromosome-level genome assemblies. **(C)** N50 values (Mb) as calculated by assembly-stats (Github: https://github.com/sanger-pathogens/assembly-stats), with asterisks marking purged genome assemblies. **(D)** BUSCO gene (microsporidia_odb10) completeness percentage, marked in green for single-copy genes, and beige for duplicated genes. **(E)** Transposable element percentage as predicted by RepeatModeler and RepeatMasker [78,79], marked in burgundy for retroelements, peach for DNA transposons, and blue for rolling circles. Neop.: Neopereziida; Or. Lin.: Orphan Lineage. The data underlying A can be found in S1 Text. The data underlying B, C, D, and E be found in S1 Table. The figure was generated using ToyTree [73], and manually annotated using InkScape (version 1.2.2).
(ZIP)

**S4 Fig. Comparison of whole-genome phylogeny species delineation thresholds and individual gene phylogeny branch length distribution species delineation thresholds.** The approach we presented in the main text relies on branch lengths derived from the whole-genome phylogeny in Fig 2 (i.e., a concatenated supermatrix of genes). We re-estimated same-species branch length thresholds for each gene. For each gene, we used the distribution of branch lengths between genomes known to belong to the same species, and measured each distribution's mean and 95th percentile. The upper threshold was then set by retrieving the highest observed 95th percentile (orange dashed line) and the highest observed mean (magenta dashed line). While the percentage of genes exceeding each threshold varies for each genome, they are relatively consistent, and lead to the same OTU assignment and the same conclusions when investigating tetraploid species. ilAceEphe1.μ still stands out as possessing more genes which exceed the same-species threshold (no matter what threshold was used) than other genomes. The figure was generated using Matplotlib [92], and manually annotated using InkScape (version 1.2.2).
(PNG)

**S5 Fig. Relationship between whole-genome phylogeny species delineation thresholds and individual gene phylogeny branch length distribution species delineation thresholds.** We compared our two gene-based metrics (highest 95th percentile and highest mean of branch length distributions of individual gene trees for genomes known to belong to the same species) to the whole-genome-based metric (highest branch length observed between any two same species genomes). We found the relationship between them to be consistent and linear, in line with the fact that they lead to the same conclusions. The figure was generated using Matplotlib [92], and manually annotated using InkScape (version 1.2.2).
(PNG)

**S6 Fig. Tetraploid ilAceEphe1.μ is uneven and rearranged.** The number of BUSCO genes found in X haplotypes, along with their total copy number. idChiSpeb1.μ is an even tetraploid, so nearly all its BUSCO genes are in 4 copies, distributed across 4 haplotypes. On the other hand, ilAceEphe1.μ is an uneven tetraploid. The majority of its BUSCO genes are in less than 4 copies, and they are not evenly distributed across its haplotypes. For instance, some BUSCO genes occur in 3 copies present only in a single haplotype. The figure was generated using gerbil (Github: https://github.com/Amjad-Khalaf/gerbil), and manually annotated using InkScape (version 1.2.2).
(PNG)

**S7 Fig. Phylogeny used by Syngraph, with its internal node labeling.** Each node is labeled with its Syngraph name in a gray box. Yellow boxes indicate the number of chromosomes each genome possesses, and blue boxes indicate the number of chromosomes which possess BUSCO gene markers. The figure was generated using ToyTree [73], and manually annotated using InkScape (version 1.2.2).
(PNG)

**S8 Fig. Number of chromosomes inferred at each node is highly variable.** The number of chromosomes inferred for each node, and the total number of BUSCO genes assigned to a chromosome for each "$m$." "$m$" is the parameter in Syngraph to determine the minimum number of genes needed to travel together for the event to be counted as a rearrangement. For example, if $m = 3$, only rearrangements involving 3 or more genes will be counted. Deep nodes are highly variable and their karyotype (and thus the number of rearrangements that have occurred along each branch) cannot be estimated reliably. See S7 Fig for node labels on the phylogeny. The figure was generated using Matplotlib [92], and manually annotated using InkScape (version 1.2.2).
(PNG)

**S9 Fig. T-SNE plot depicting BUSCO linkage groups across the microsporidian phylogeny.** Each point represents a BUSCO gene, positioned based on its co-occurrence profile across the chromosome-level microsporidian genomes. Distances between points reflect similarities in co-occurrence. Points are coloured by their assigned chromosome in *Anotonspora locustae*. This disorganized pattern illustrates that the rate of rearrangement is too high for a reliable complete reconstruction of putative ancestral linkage groups. The large-scale patterns are influenced by more densely sampled taxa, see S10 Fig. The data underlying this figure can be found in File Collection 5 at https://doi.org/10.5281/zenodo.17251512. The figure was generated using Scikit-learn [142,143] and Matplotlib [92], and manually annotated using InkScape (version 1.2.2).
(PNG)

**S10 Fig. T-SNE plot depicting BUSCO linkage groups across the microsporidian phylogeny, highlighting clustering influence by more densely sampled taxa.** Each point represents a BUSCO gene, positioned based on its co-occurrence profile across the chromosome-level microsporidian genomes. Distances between points reflect similarities in co-occurrence. Points are coloured by their assigned chromosome in *Encephalitozoon cuniculi*. This disorganized pattern illustrates that the rate of rearrangement is too high for a reliable complete reconstruction of

putative ancestral linkage groups. The large-scale patterns are influenced by more densely sampled taxa, such as *Encephalitozoon cuniculi*. The data underlying this figure can be found in File Collection 5 at https://doi.org/10.5281/zenodo.17251512. The figure was generated using Scikit-learn [142,143] and Matplotlib [92], and manually annotated using InkScape (version 1.2.2).
(PNG)

**S11 Fig. Synteny plots of chromosomal microsporidian genome assemblies.** Genome-wide synteny plots of all available chromosomal microsporidian genome assemblies. Each line represents a single-copy BUSCO (microsporidia_odb10) [76]. BUSCOs are painted by their chromosomal position in *A. locustae*. The data underlying this figure can be found in File Collection 5 at https://doi.org/10.5281/zenodo.17251512. Figure was generated by using ribbon plot scripts from https://github.com/conchoecia/odp [109] and ToyTree [73], and manually annotated using InkScape (version 1.2.2).
(PNG)

**S12 Fig. Synteny plots of chromosomal microsporidian genome assemblies.** Genome-wide synteny plots of all available chromosomal microsporidian genome assemblies. Each line represents a single-copy BUSCO (microsporidia_odb10) [76]. BUSCOs are painted by their chromosomal position in *H. tvaerminnensis*. Figure was generated by using ribbon plot scripts from https://github.com/conchoecia/odp [109] and ToyTree [73], and manually annotated using InkScape (version 1.2.2).
(PNG)

## Acknowledgments

We thank Dr Lewis Stevens, Dr Jamie Bojko, and Dr Yuliya Y Sokolova for their insight, their kindness, and generosity with their time in discussing these results with us over the last year. We also warmly thank our colleagues at the Tree of Life Programme, Wellcome Sanger Institute for their support, and comradery.

## Author contributions

**Conceptualization:** Amjad Khalaf.

**Data curation:** Amjad Khalaf.

**Formal analysis:** Amjad Khalaf.

**Funding acquisition:** Mark Blaxter, Mara K. N. Lawniczak.

**Investigation:** Amjad Khalaf.

**Methodology:** Amjad Khalaf, Chenxi Zhou, Claudia C. Weber, Emmelien Vancaester, Ying Sims, Alex Makunin, Kamil S. Jaron, Mark Blaxter, Mara K. N. Lawniczak.

**Resources:** Mark Blaxter, Mara K. N. Lawniczak.

**Software:** Amjad Khalaf, Ying Sims, Shane A. McCarthy.

**Supervision:** Claudia C. Weber, Mark Blaxter, Mara K. N. Lawniczak.

**Validation:** Thomas C. Mathers, Dominic E. Absolon, Jonathan M. D. Wood.

**Visualization:** Amjad Khalaf.

**Writing – original draft:** Amjad Khalaf.

**Writing – review & editing:** Amjad Khalaf, Chenxi Zhou, Claudia C. Weber, Emmelien Vancaester, Alex Makunin, Mark Blaxter, Mara K. N. Lawniczak.

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
