## [Editor Report · Decision Letter 0]

20 May 2025

Dear Dr Khalaf,

Thank you for submitting your manuscript entitled "Forty New Genomes Shed Light on Sexual Reproduction and the Origin of Tetraploidy in Microsporidia" for consideration as a Research Article by PLOS Biology.

Your manuscript has now been evaluated by the PLOS Biology editorial staff, as well as by an academic editor with relevant expertise, and I'm writing to let you know that we would like to send your submission out for external peer review.

Once your full submission is complete, your paper will undergo a series of checks in preparation for peer review. After your manuscript has passed the checks it will be sent out for review. To provide the metadata for your submission, please Login to Editorial Manager (https://www.editorialmanager.com/pbiology) within two working days, i.e. by May 22 2025 11:59PM.

Kind regards,

Roli Roberts

Roland Roberts, PhD

Senior Editor

PLOS Biology

rroberts@plos.org

---

## [Decision Letter · Decision Letter 1]

25 Jun 2025

Dear Dr Khalaf,

Thank you for your patience while your manuscript "Forty New Genomes Shed Light on Sexual Reproduction and the Origin of Tetraploidy in Microsporidia" was peer-reviewed at PLOS Biology. It has now been evaluated by the PLOS Biology editors, an Academic Editor with relevant expertise, and by three independent reviewers.

You'll see that reviewer #1 is positive about the study, but thinks that the manuscript needs significant streamlining, especially for our broader readership. S/he also wants better treatment of the prior literature around sex in microsporidia, including population genetic support for this claim. There are also some presentational (to include a table of the genomes’ properties) and methodological (to justify the use of prokaryotic annotation pipeline) requests. Reviewer #2 is also positive, but wants improved analysis and presentation of the phylogeny, expressed scepticism about the number of genes in some species, and has multiple queries about the methods (including how you excluded the co-occurrence of multiple microsporidia in a single host). Reviewer #3 starts very positive, but then becomes more critical, raising concerns about inferring ploidy without cytological data, and the overall lack of methodological clarity. Like reviewer #1, s/he also thinks that the paper needs streamlining.

In light of the reviews, which you will find at the end of this email, we would like to invite you to revise the work to thoroughly address the reviewers' reports.

Given the extent of revision needed, we cannot make a decision about publication until we have seen the revised manuscript and your response to the reviewers' comments. Your revised manuscript is likely to be sent for further evaluation by all or a subset of the reviewers.

**IMPORTANT - SUBMITTING YOUR REVISION**

*Re-submission Checklist*

*Published Peer Review*

*PLOS Data Policy*

*Blot and Gel Data Policy*

Sincerely,

Roli Roberts

Roland Roberts, PhD

Senior Editor

PLOS Biology

rroberts@plos.org

REVIEWERS' COMMENTS:

Reviewer #1:

The paper by Khalaf and colleagues provides a comprehensive analysis of 40 microsporidian genomes derived from long-read sequencing datasets of infected animal hosts. The study is robust, with results that are generally well-supported by the provided data and analyses. Below, I offer suggestions to improve the paper's structure, readability, and scientific rigor, along with specific technical comments.

Note that the absence of line numbers and the placement of figure legends (separated from figures and embedded mid-text) made the review process unnecessarily challenging, and I recommend addressing this for clarity.

General Comments:

* Streamline Content for Conciseness and Broader Appeal: While the dataset is extensive, the paper could be significantly shortened by consolidating analyses addressing similar questions. For instance, the phylogenetic and species definition analyses could be combined into a single, concise paragraph, as they are closely related. Similarly, analyses of inter-species versus intra-haplotype rearrangement/recombination both explore genomic diversity in microsporidia and should be merged into one section. Reducing overly detailed discussions of niche interest (e.g., specific phylogenetic nuances) would enhance accessibility for the broader readership of PLoS Biology.

* Clarify "Compartments" Terminology: The use of "compartments" in the context of eukaryotic genomes typically refers to A/B compartments—genomic regions with distinct gene/repeat enrichments and conformational differences tied to the cell cycle. The Hi-C data presented suggests compartmentalization, but it's unclear whether these reflect standard A/B compartments, as seen in other eukaryotes, including symbiotic or pathogenic fungi. The authors should investigate and explicitly compare their findings to known A/B compartmentalization to avoid confusion.

* Acknowledge Prior Hypotheses on Microsporidian Sex and Ploidy: The paper's major claims, such as the proposed sexual reproduction model and diploid/tetraploid cycles, build on earlier hypotheses that are not cited. For example, Pelin et al. (2016) and Corradi (2015) discuss diploid-tetraploid cycles and potential karyogamy in microsporidia, suggesting mechanisms for genetic diversity. These seminal works must be referenced to contextualize the current findings. The authors should clarify whether their data supports, refines, or challenges these hypotheses. Additionally, claims of sexual reproduction should be substantiated with population genetics analyses, which are absent here but have been explored elsewhere (e.g., in studies not cited in this manuscript).

Specific Comments:

* Page 7, Compartmentalization: The statement, "We show that one tetraploid genome is organised into two compartments, likely the nuclei of the diplokaryon," requires clarification. Do these Hi-C-defined compartments show differential enrichment in transposable elements (TEs), BUSCO genes, or other features, as expected in A/B compartments? The authors should compare their findings to known eukaryotic compartmentalization to ensure accurate terminology.

* Page 9, Genome Data Accessibility: The paper lacks a clear summary of the 40 genomes' characteristics. I recommend including a table detailing genome statistics (e.g., size, scaffold number, N50), sequencing methods (e.g., Hi-C, telomere resolution), and quality metrics. Additionally, explain why Hi-C was applied to some genomes but not others and outline the sequencing-to-scaffolding pipeline to provide context for methodological choices.

* Page 11, Figure and Legend Placement: Embedding figure legends mid-text, with figures placed at the end, hinders readability. Figures and their legends should be integrated within the main text at relevant points to facilitate review and comprehension.

* Page 12, Annotation Pipeline: The bacterial annotation pipeline is surprising given microsporidia's eukaryotic nature. Although introns are rare and small, a eukaryotic annotation pipeline is standard and should be employed. Justify the use of a bacterial pipeline or revise the methodology.

* Page 12, Transposable Elements and rRNA: The paper notes correlations between retroelement, DNA transposon, and helitron loads with genome size but lacks detail on their genomic distribution. Are TEs or rRNA operons enriched in specific chromosomal regions, such as subtelomeres, as observed in Encephalitozoon? This should be explored and discussed.

* Page 13, Species Definitions: Defining species based on branch length is problematic, as it risks oversimplification (e.g., analogous to grouping marsupials and mammals as conspecific). Species delimitation in microsporidia should incorporate population genetics approaches to assess genetic exchange, as asexual lineages may not align with traditional species concepts. The authors should acknowledge these limitations and consider integrating population genetics data to support taxonomic claims.

* Page 15, Nomenclature: Designations like "idChiSpeb1.μ" are opaque and confusing. If a species name exists (e.g., for the sample from Chironomus sp.), it should be used consistently to improve clarity.

Reviewer #2:

Microsporidia are a large group of intracellular parasites that infect many species of animals. Their genomes have been of interest as they have the smallest known eukaryotic genomes, but their genomic structure has not been thoroughly investigated due to only have a few species with chromosomal level assemblies. Here the authors take advantage of genome reads from serendipitously infected insects to assemble 40 genomes, representing 17 species. The authors use these genome assemblies to investigate ploidy, genome structure, and provide evidence for sexual reproduction. Overall, this is an interesting paper and the high quality microsporidia genome assemblies will be a useful resource.

Major points:

1. The authors' claim "They allowed us to revise microsporidian phylogeny, notably changing the position of Glugeida to the sister group of the ancestor of the 'orphan lineage and Amblyosporida" is unsupported by the provided data. There needs to be support values provided for the nodes of the tree in figure 2 and ideally more than one phylogenetic approach would be used. The authors use terms like "clustering robustly" and "We confidently placed" , but it is unclear what these descriptions are supported by. The authors choice to use metchnikoviids as the root is not explained, as most groups show that Mitosporidium daphnae is sister to the metchnikoviids and microsporidia. The nodes are also very difficult to see as they are compressed. I would recommend creating a separate version of this tree for the main figure that is just species and not strains. The full tree can then be a supplemental figure.

2. The number of genes predicted which are shown in figure 2 seem unreasonably high. Some species of microsporidia are reported to have less than 2000 protein coding genes, so the reporting at least 10 species having over 20,000 proteins is unexpected. What is the explanation for this? Is this different variants for the same gene being predicted multiple times? Or is this just very short proteins? Other methods besides Prokka that have previously been used on microsporidia genomes should be investigated (see the following paper for examples: DOI: 10.1111/jeu.13038.) The authors state that "no functional annotations were required in this piece of work", but gene prediction affects analysis such as figure 4a and figure 8 which some species have 100 gene pairs vs 1000 gene pairs. Also for these genomes to be useful resource, the annotations along with the genome assemblies should be deposited in a central repository such as NCBI.

3. There are very high levels of duplicated buscos (up to close to 100%) in some assemblies. Is this just because some of the assemblies are haploid resolved? This needs to be explained.

4. How the paper deals with purged genomes is confusing. A better explanation of how and why genomes were purged would be helpful to add.

5. The authors do not appear to take any steps to confirm that the microsporidia reads from their samples come from a single species. Co-occurrence of different microsporidia species in mosquitos has been shown to be ~10% (https://doi.org/10.1111/1755-0998.13205). For the non-chromosomal level genomes, some criteria needs to be used to determine whether the microsporidia reads in each sample are from the same species.

6. What criteria is used to classify ploidy in an assembly needs to be defined. For example, why is the ploidy of iyOphElle1.µ, not determined, as this is a chromosome assembly?

7. The number of genome assemblies that were screened using blobtools for microsporidia reads is not mentioned and should be.

Minor points:

1. I would recommend against using µ as to name the novel microsporidia species genomes. There will certainly be instances where others will use these genomes in applications where Greek letters are not able to be used, and this could lead to confusion. This is a known issue in biology nomenclature and see the following paper for this recommendation in the context of gene names: DOI: 10.1006/geno.2002.6748 . I would suggest using some other abbreviation such "m", "micro", or "ms" that makes the same point using the Latin alphabet.

2. Methods are lacking for the primers and genes that were used to identify individuals as microsporidia positive.

3. In figure 1A "number of microsporidia genomes" would be more appropriate as the X-axis label and moved underneath the X-axis.

4. Several acronyms and abbreviations are used that are not defined such as MFP, Asm., and OUT.

5. The paper often uses a lot of jargon that makes the paper less accessible, such as "Operational taxonomic unit classification of species suggests autotetraploidy in Microsporidia". It would be helpful to revise the paper with an eye towards clarity and simplicity that would help non-specialists be able to understand.

6. The scale in figure 3 is too compressed see the branch lengths of .15 that the authors point as delineating species. It would be helpful to use a different scale that allows this level of differences to be visualized.

7. What is being a considered an OTU in this paper not explained.

8. The text in figure 4A is too small to read and needs to be increased to at least 8 pt.

9. In figure 5 the clade names at the bottom are upside down and should be flipped so they can be more easily read.

10. A scale needs to be shown for figure 7.

Reviewer #3:

Khalaf et al. report on 40 new Microsporidia genomes that were sequenced incidentally as part of the Darwin Tree of Life project. The authors generated rather intact, complete genomes of these parasites and used Hi-C to assess contacts between homologous chromosomes in a genome. The authors use the data to address a large number of questions in Microsporidia biology, such as phylogeny, synteny, and most interestingly the evolution of tetraploidy. The authors show that tetraploids are autopolyploids rather than allopolyploids. A novel hypothesis that the two nuclei inside diplokaryotic cells are actually in some cases tetraploids divided into diploid nuclear compartments is proposed. Overall, this study provides new insights into the biology of these fascinating parasites and is easy to read.

However, I have a few major criticisms that dampen my enthusiasm. Firstly, this paper really rests on really accurate estimation of ploidy, which is tricky without cytology. For example, it is impossible to distinguish a haploid from a homozygous diploid or tetraploid. The presentation of how ploidy was assigned to these genomes was not sufficient to evaluate how robust the assignment was. Some of this comes up in Figure 9, where there is a mixed signal which the authors ascribe to segmental duplication. To be clear, I don't think the authors did a poor job in their assignments, I just think they need to be clearer about how they were done and whether there was any ambiguity. A second issue relates to the Hi-C data. Here, the authors could do a better job in walking us through what is happening. The figure itself is not self explanatory, and while their hypothesis is interesting, it's not clear how well it is supported by the data. How much uncertainty is there in this diplokaryotic model? Third, the authors run through many different analyses that are interesting, but again require really solid ploidy estimation and phasing/assembly of data into homeologues, such as the bizarre finding of an uneven and rearranged tetraploid.

I have few other general comments, but again the paper is pretty cleanly written.

1. The authors should probably streamline some of the Results/presentation. For example Figure 1B is not a big result and there is not much they can say about it. Is it worth even mentioning?

2. A new way of defining species is presented using amino acid substitutions per site. This seems more complicated that the recent analysis by Albuquerque and Haag, doi: 10.1111/jeu.12944 using ANI. I'm not sure the field needs a more complicated method that involves more variation in molecular rates.

3. Figure 4 only uses tetraploids. However, if allopolyploidy occurred between two species that were diploid but homozygous, it could look the same as a diploid.

4. Without evidence of deep coalescence between homeologues, is the section "No evidence of recent rediploidisation in Microsporidia" likely to find anything?

5. Figure 12 could use a little polishing as it shows meiosis yielding two products and gametes undergoing mitosis. These simplifications could end up confusing readers.

---

## [Decision Letter · Decision Letter 2]

29 Sep 2025

Dear Dr Khalaf,

Thank you for your patience while we considered your revised manuscript "Forty New Genomes Shed Light on Sexual Reproduction and the Origin of Tetraploidy in Microsporidia" for publication as a Research Article at PLOS Biology. This revised version of your manuscript has been evaluated by the PLOS Biology editors, the Academic Editor, and the original reviewers.

Based on the reviews, we are likely to accept this manuscript for publication, provided you satisfactorily address the remaining points raised by the reviewers and the following data and other policy-related requests.

IMPORTANT - please attend to the following:

a) Please address the remaining points from the reviewers...

b) ...of which the Academic Editor said "I think that reviewer #2 has a valid point that it can be hard to discern if these are clonal vs polyclonal infections. I don't think it takes away from their major conclusions to add this as a caveat given that extent of analysis. As for the protein predictions, I think you can ask that they also address this and ensure that the methods they have used are sufficiently documented that this can be repeated by others working in the field." I've included that in case it's helpful.

c) Please address my Data Policy requests below; specifically, we need you to supply the numerical values underlying Figs 1, 2ABCDE, 3AB, 4, 5, 6A, 7, 8ABC, 9AB, and all of the Supp Figs, either as a supplementary data file or as a permanent DOI’d deposition. I note that you already have an associated Zenodo deposition (https://doi.org/10.5281/zenodo.15364388). Please could you confirm whether the data and code in this deposition are sufficient to recreate the Figures (main and supplementary)?

d) Please cite the location of the data clearly in all relevant main and supplementary Figure legends, e.g. “The data underlying this Figure can be found in S1 Data” or “The data underlying this Figure can be found in https://zenodo.org/records/15364388”

e) I note that (unlike in earlier versions) the supplementary Figs and Tables are currently only accessible as part of very large zipped folders – please submit these as separate files, and include their legends in the manuscript file. It’s fine for the other files (non-Fig, non-Table) to simply be provided in the Zenodo deposition.

f) Please make any custom code available, either as a supplementary file or as part of your Zenodo deposition.

g) Please include the URLs of your funders in the Financial Disclosure statement.

We expect to receive your revised manuscript within two weeks.

*Published Peer Review History*

*Press*

Sincerely,

Roli Roberts

Roland Roberts, PhD

Senior Editor

rroberts@plos.org

PLOS Biology

DATA POLICY:

Regardless of the method selected, please ensure that you provide the individual numerical values that underlie the summary data displayed in the following figure panels as they are essential for readers to assess your analysis and to reproduce it: Figs 1, 2ABCDE, 3AB, 4, 5, 6A, 7, 8ABC, 9AB, and all of the Supp Figs. NOTE: the numerical data provided should include all replicates AND the way in which the plotted mean and errors were derived (it should not present only the mean/average values).

CODE POLICY

DATA NOT SHOWN?

REVIEWERS' COMMENTS:

Reviewer #1:

No comments.

Reviewer #2:

Although the authors do address some of my concerns there are some that they don't. My biggest remaining concern is the lack of protein predictions and annotations. Although the authors don't use genome wide protein predictions them in their analysis, a large value of a paper that assembles novel genomes, is that the genomes are used as a resource, and the prediction and annotation of proteins is a large part of that. I also don't understand the data on the protein prediction numbers the authors provide. It seems like AUGUSTUS for many of the genomes is predicting 0 proteins. Presumably for their BUSCO analysis they are relying on AUGUSTUS predictions, so its unclear to me why their predictions are so low. My other remaining concerns are listed below.

1.The authors claim that the genomes are clonal infections but they still don't provide convincing data to support that. Their marker scan approach is relying on 1000 bp long sequences (this is from the cited paper, the authors should include this information in their methods) and if only 10 percent of the genome might be sequenced, this might result in no ribosomal sequence or only a fragmented ribosomal sequence. Their genome scan results also likely wouldn't be able to detect if only 10 percent of the genome was from a different species. Their blob tools analysis could potentially deal with a partial cooccurring genome. The authors state: " A mixed infection of multiple microsporidians would display a distinct pattern in the BlobToolKit plots - especially given that the two (or more) microsporidians would likely differ in read depth (because of different levels of infection)," Here are a couple of examples of non-chromosomal genomes and the range of coverages that they see. Do the authors have evidence that a co-infection that is at 10% of the main infection would not have any contigs included within the low range of coverage?

iyAmbProt1.µ 2.219 - 41.448

ihCicViri2.µ 21.3 - 345.6

Although these genomes may indeed be from single clonal infections, I do not believe the authors have provided evidence to support their new claim "we determined that all of our forty microsporidian genomes are derived from single, clonal infections.""

I do think this is a hard problem and the authors do not necessarily have to solve it, but If the authors are not going to convincingly address this, adding a sentence to the discussion or methods about the limitation of analysing microsporidia genomes from metagenomic data would be helpful.

2. The authors are still rooting their tree based on Metchnikovella, without an explanation why. This doesn't effect their results, but displaying the tree this way will be confusing as Mitosporidium is considered the outgroup to metchnikovella and the microsporidia.

Reviewer #3:

I have evaluated the revision, and I think the authors have done a decent job of addressing the previous comments. I note that the responses are somewhat hard to follow because the line numbers do not jive with the rebuttal.

I think the authors should keep in mind that actually it is very easy to have a situation where you have a completely homozygous diploid. There are many eukaryotes where haploid selfing occurs and generates completely homozygous diploids. They do acknowledge that a completely homozygous diploid is impossible to differentiate from a completely homozygous tetraploid or haploid. I also stick behind my statement about ANI, but we can disagree on this opinion and this doesn't influence the results in this paper.

Congratulations to the authors.

---

## [Editor Report · Decision Letter 3]

3 Oct 2025

Dear Dr Khalaf,

Thank you for the submission of your revised Research Article "Forty New Genomes Shed Light on Sexual Reproduction and the Origin of Tetraploidy in Microsporidia" for publication in PLOS Biology. On behalf of my colleagues and the Academic Editor, Joseph Heitman, I'm pleased to say that we can in principle accept your manuscript for publication, provided you address any remaining formatting and reporting issues. These will be detailed in an email you should receive within 2-3 business days from our colleagues in the journal operations team; no action is required from you until then. Please note that we will not be able to formally accept your manuscript and schedule it for publication until you have completed any requested changes.

Sincerely,

Roli Roberts

Senior Editor

PLOS Biology

rroberts@plos.org